# Compositional Foundation Models for Hierarchical Planning

**Anurag Ajay**[*†§], **Seungwook Han** [*†‡§] **Yilun Du** [*§], **Shuang Li** [§],
**Abhi Gupta** [§], **Tommi Jaakkola** [§], **Josh Tenenbaum** [§], **Leslie Kaelbling** [§],
**Akash Srivastava** [‡], **Pulkit Agrawal** [†§]
Improbable AI Lab[†]  MIT-IBM Watson AI Lab[‡]  Massachusetts Institute Technology[§]
https://hierarchical-planning-foundation-model.github.io/

## Abstract

To make effective decisions in novel environments with long-horizon goals, it is crucial to engage in hierarchical reasoning across spatial and temporal scales. This entails planning abstract subgoal sequences, visually reasoning about the underlying plans, and executing actions in accordance with the devised plan through visual-motor control. We propose *Compositional Foundation Models for Hierarchical Planning* (HiP), a foundation model which leverages multiple *expert* foundation model trained on language, vision and action data *individually* jointly together to solve long-horizon tasks. We use a large language model to construct symbolic plans that are grounded in the environment through a large video diffusion model. Generated video plans are then grounded to visual-motor control, through an inverse dynamics model that infers actions from generated videos. To enable effective reasoning within this hierarchy, we enforce consistency between the models via *iterative refinement*. We illustrate the efficacy and adaptability of our approach in three different long-horizon table-top manipulation tasks.

## 1 Introduction

Consider the task of making a cup of tea in an unfamiliar house. To successfully execute this task, an effective approach is to reason hierarchically at multiple levels: an abstract level, *e.g.* the high level steps needed to heat up the tea, a concrete geometric level *e.g.*, how we should physically navigate to and in the kitchen, and at a control level *e.g.* how we should actuate our joints to lift a cup. It is further important that reasoning at each level is self-consistent with each other – an abstract plan to look in cabinets for tea kettles must also be physically plausible at the geometric level and executable given the actuations we are capable of. In this paper, we explore how we can create agents capable of solving novel long-horizon tasks which require hierarchical reasoning.

Large "foundation models" have become a dominant paradigm in solving tasks in natural language processing [36, 47, 7], computer vision [26], and mathematical reasoning [27]. In line with this paradigm, a question of broad interest is to develop a "foundation model" that can solve novel and long-horizon decision-making tasks. Some prior works [39, 6] collected paired visual, language and action data and trained a monolithic neural network for solving long-horizon tasks. However, collecting paired visual, language and action data is expensive and hard to scale up. Another line of prior works [10, 28] finetune large language models (LLM) on both visual and language inputs using task-specific robot demonstrations. This is problematic because, unlike the abundance of text on the Internet, paired vision and language robotics demonstrations are not readily available and are expensive to collect. Furthermore, finetuning high-performing language models, such as GPT3.5/4 [37, 36] and PaLM [7], is currently impossible, as the model weights are not open-sourced.

---

[*] denotes equal contribution. Authors are also affiliated with Computer Science and Artificial Laboratory (CSAIL). Correspondence to aajay@mit.edu, swhan@mit.edu and yilundu@mit.edu.

37th Conference on Neural Information Processing Systems (NeurIPS 2023).

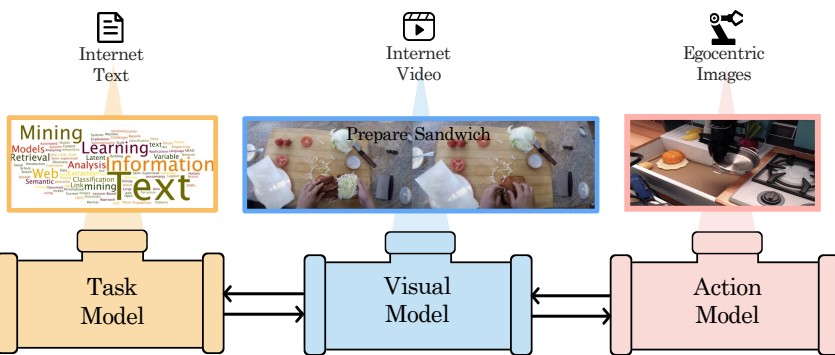

Figure 1: **Compositional Foundation Models for Hierarchical Planning.** `HiP` uses a task model, represented using a LLM, to create an abstract plan, a visual model, represented using a video model, to generate an image trajectory plan, and an ego-centric action model to infer actions from the image trajectory.

The key characteristic of the foundation model is that solving a new task or adapting to a new environment is possible with much less data compared to training from scratch for that task or domain. Instead of building a foundation model for long-term planning by collecting paired language-vision-action data, in this work we seek a scalable alternative – can we reduce the need for a costly and tedious process of collecting paired data across three modalities and yet be relatively efficient at solving new planning tasks? We propose *Compositional Foundation Models for Hierarchical Planning* (`HiP`), a foundation model that is a composition of different *expert* models trained on language, vision, and action data *individually*. Because these models are trained individually, the data requirements for constructing the foundation models are substantially reduced (Figure 1). Given an abstract language instruction describing the desired task, `HiP` uses a large language model to find a sequence of sub-tasks (i.e., planning). `HiP` then uses a large video diffusion model to capture geometric and physical information about the world and generates a more detailed plan in form of an observation-only trajectory. Finally, `HiP` uses a large pre-trained inverse model that maps a sequence of ego-centric images into actions. The compositional design choice for decision-making allows separate models to reason at different levels of the hierarchy, and jointly make expert decisions without the need for ever collecting expensive paired decision-making data across modalities.

Given three models trained independently, they can produce inconsistent outputs that can lead to overall planning failure. For instance, a naïve approach for composing models is to take the maximum-likelihood output at each stage. However, a step of plan which is high likelihood under one model, *i.e.* looking for a tea kettle in a cabinet may have zero likelihood under a seperate model, *i.e.* if there is no cabinet in the house. It is instead important to sample a plan that jointly maximizes likelihood across every expert model. To create consistent plans across our disparate models, we propose an *iterative refinement* mechanism to ensure consistency using feedback from the downstream models [28]. At each step of the language model's generative process, intermediate feedback from a likelihood estimator conditioned on an image of the current state is incorporated into the output distribution. Similarly, at each step of the video model generation, intermediate feedback from the action model refines video generation. This *iterative refinement procedure* promotes consensus among the different models and thereby enables hierarchically consistent plans that are both responsive to the goal and executable given the current state and agent. Our proposed iterative refinement approach is computationally efficient to train, as it does not require any large model finetuning. Furthermore, we do not require access to the model's weights and our approach works with any models that offer only input and output API access.

In summary, we propose a compositional foundation model for hierarchical planning that leverages a composition of foundation models, learned separately on different modalities of Internet and ego-centric robotics data, to construct long-horizon plans. We demonstrate promising results on three long-horizon tabletop manipulation environments.

## 2   Compostional Foundation Models for Hierarchical Planning

We propose `HiP`, a foundation model that decomposes the problem of generating action trajectories for long-horizon tasks specified by a language goal $g$ into three levels of hierarchy: **(1)** Task planning – inferring a language subgoal $w_i$ conditioned on observation $x_{i,1}$ and language goal $g$; **(2)** Visual

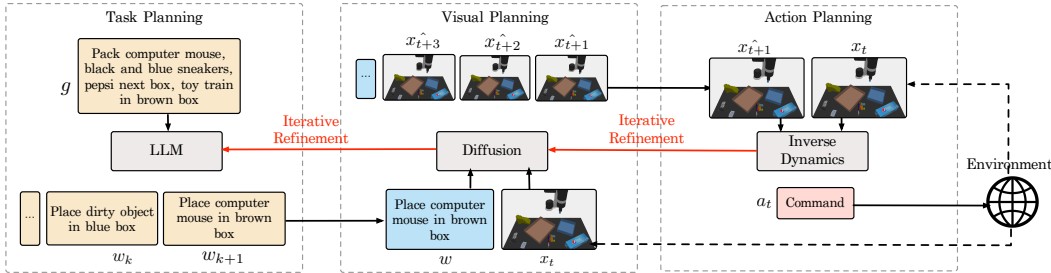

Figure 2: **Planning with** `HiP`. Given a language goal $g$ and current observation $x_t$, LLM generates next subgoal $w$ with feedback from a visual plausibility model. Then, Diffusion uses observation $x_t$ and subgoal $w$ to generate observation trajectory $\tau_x$ with feedback from an action feasibility model. Finally, action planning uses inverse dynamics to generate action $a_t$ from current $x_t$ and generated observation $\hat{x_{t+1}}$ (action planning).

planning – generating a physically plausible plan as a sequence of image trajectories $\tau_x^i = \{x_{i,1:T}\}$, one for each given language subgoal $w_i$ and observation at first timestep $x_{i,1}$; **(3)** Action planning – inferring a sequence of action trajectories $\tau_a^i = \{a_{i,1:T-1}\}$ from the image trajectories $\tau_x^i$ executing the plan. Figure 2 illustrates the model architecture and a pseudocode is provided in Algorithm 1.

Let $p_\Theta$ model this hierarchical decision-making process. Given our three levels of hierarchy, $p_\Theta$ can be factorized into the following: task distribution $p_\theta$, visual distribution $p_\phi$, and action distribution $p_\psi$. The distribution over plans, conditioned on the goal and an image of the initial state, can be written under the Markov assumption as:

$$p_\Theta(W, \{\tau_x^i\}, \{\tau_a^i\}|g, x_{1,1}) = \underbrace{\left(\prod_{i=1}^{N} p_\theta(w_i|g)\right)}_{\text{task planning}} \underbrace{\left(\prod_{i=1}^{N} p_\phi(\tau_x^i|w_i, x_{i,1})\right)}_{\text{visual planning}} \underbrace{\left(\prod_{i=1}^{N}\prod_{t=1}^{T-1} p_\psi(a_{i,t}|x_{i,t}, x_{i,t+1})\right)}_{\text{action planning}} \quad (1)$$

We seek to find action trajectories $\tau_a^i$, image trajectories $\tau_x^i$ and subgoals $W = \{w_i\}$ which maximize the above likelihood. Please see Appendix A for a derivation of this factorization. In the following sub-sections, we describe the form of each of these components, how they are trained, and how they are used to infer a final plan for completing the long-horizon task.

## 2.1 Task Planning via Large Language Models

Given a task specified in language $g$ and the current observation $x_{i,1}$, we use a pretrained LLM as the task planner, which decomposes the goal into a sequence of subgoals. The LLM aims to infer the next subgoal $w_i$ given a goal $g$ and models the distribution $p_{\text{LLM}}(w_i|g)$. As the language model is trained on a vast amount of data on the Internet, it captures powerful semantic priors on what steps should be taken to accomplish a particular task. To adapt the LLM to obtain a subgoal sequence relevant to our task, we prompt it with some examples of domain specific data consisting of high-level goals paired with desirable subgoal sequences.

However, directly sampling subgoals using a LLM can lead to samples that are inconsistent with the overall joint distribution in Eqn (1), as the subgoal $w_i$ not only affects the marginal likelihood of task planning but also the downstream likelihoods of the visual planning model. Consider the example in Figure 2 where the agent is tasked with packing computer mouse, black and blue sneakers, pepsi box and toy train in brown box. Let's say the computer mouse is already in the brown box. While the subgoal of placing computer mouse in brown box has high-likelihood under task model $p_\theta(w_i|g)$, the resulting observation trajectory generated by visual model $p_\phi(\tau_x^i|w_i, x_{i,1})$ will have a low-likelihood under $p_\phi$ given the subgoal is already completed. Next, we describe how we use iterative refinement to capture this dependency between language decoding and visual planning to sample from Eqn (1).

**Consistency with Visual Planning**   To ensure that we sample subgoal $w_i$ that maximizes the joint distribution in Eqn (1), we should sample a subgoal that maximizes the following joint likelihood

$$w_i^* = \arg\max_{w_i} p_{\text{LLM}}(w_i|g)p_\phi(\tau_x^i|w_i, x_{i,1}), \quad (2)$$

i.e. a likelihood that maximizes both conditional subgoal generation likelihood from a LLM and the likelihood of sampled videos $\tau_x^i$ given the language instruction and current image $x_{i,1}$. One way to determine the optimal subgoal $w_i^*$ is to generate multiple $w_i$ from LLM and score them using the likelihood of videos sampled from our video model $p_\phi(\tau_x^i|w_i, x_{i,1})$. However, the video generation process is computationally expensive, so we take a different approach.

---

**Algorithm 1** Decision Making with `HiP`

---

1: **Models:** Large language model $p_{\text{LLM}}$, Subgoal classifier $f_\phi$, Noise model of diffusion $\epsilon_\phi$, Observation trajectory classifier $g_\psi$, Inverse dynamics $p_\psi$
2: **Hyperparameters:** Guidance scales $\omega, \omega'$, Denoising diffusion steps $K$
3: **Input:** Current observation $x_t$, Language goal $g$
4: # Task Planning
5: **for** $i = 1 \ldots M$ **do**
6:     Generate subgoal $w_i \sim p_{LLM}(w_i|g)$
7: **end for**
8: Collect candidate subgoals $W \leftarrow \{w_i\}_{i=1}^M$
9: # Iterative Refinement from Visual Planning
10: $w \leftarrow \arg\max_w f_\phi(x_t, W, g)$
11: # Visual Planning
12: Initialize $(\tau_x)_K \sim \mathcal{N}(0, I)$
13: **for** $k = K \ldots 1$ **do**
14:     # Iterative Refinement from Action Planning
15:     $\hat{\epsilon} \leftarrow \epsilon_\phi((\tau_x)_k, x_t, k) + \omega(\epsilon_\phi((\tau_x)_k, x_t, w, k) - \epsilon_\phi((\tau_x)_k, x_t, k)) - \omega' \nabla_{(\tau_x)_k} \log g_\psi(1|(\tau_x)_k)$
16:     $(\tau_x)_{k-1} \leftarrow \text{Denoise}((\tau_x)_k, \hat{\epsilon})$
17: **end for**
18: # Action Planning
19: Extract $(x_t, \hat{x_{t+1}})$ from $(\tau_x)_0$
20: Execute $a_t \leftarrow p_\psi(a_t|x_t, \hat{x_{t+1}})$

---

The likelihood of video generation $p_\phi(\tau_x^i|w_i, x_{i,1})$ primarily corresponds to the feasibility of a language subgoal $w_i$ with respect to the initial image $x_{i,1}$. Thus an approximation of Eqn (2) is to directly optimize the conditional density

$$w_i^* = \arg\max_{w_i} p(w_i|g, x_{i,1}). \tag{3}$$

We can rewrite Eqn (3) as

$$w_i^* = \arg\max_{w_i} \log p_{\text{LLM}}(w_i|g) + \log\left(\frac{p(x_{i,1}|w_i, g)}{p(x_{i,1}|g)}\right)$$

We estimate the density ratio $\frac{p(x_{i,1}|w_i,g)}{p(x_{i,1}|g)}$ with a multi-class classifier $f_\phi(x_{i,1}, \{w_j\}_{i=1}^M, g)$ that chooses the appropriate subgoal $w_i^*$ from candidate subgoals $\{w_j\}_{j=1}^M$ generated by the LLM. The classifier implicitly estimates the relative log likelihood estimate of $p(x_{i,1}|w_i, g)$ and use these logits to estimate the log density ratio with respect to each of the $M$ subgoals and find $w_i^*$ that maximizes the estimate [45]. We use a dataset $\mathcal{D}_{\text{classify}} \coloneqq \{x_{i,1}, g, \{w_j\}_{j=1}^M, i\}$ consisting of observation $x_{i,1}$, goal $g$, candidate subgoals $\{w_j\}_{j=1}^M$ and the correct subgoal label $i$ to train $f_\phi$. For further architectural details, please refer to Appendix B.1.

## 2.2 Visual Planning with Video Generation

Upon obtaining a language subgoal $w_i$ from task planning, our visual planner generates a plausible observation trajectory $\tau_x^i$ conditioned on current observation $x_{i,1}$ and subgoal $w_i$. We use a video diffusion model for visual planning given its success in generating text-conditioned videos [20, 53]. To provide our video diffusion model with a rich prior for physically plausible motions, we pretrain it $p_\phi(\tau_x^i|w_i, x_{i,1})$ on a large-scale text-to-video dataset Ego4D [13]. We then finetune it on the task-specific video dataset $\mathcal{D}_{\text{video}} \coloneqq \{\tau_x^i, w_i\}$ consisting of observation trajectories $\tau_x^i$ satisfying subgoal $w_i$. For further architectural details, please refer to Appendix B.2.

However, analogous to the consistent sampling problem in task planning, directly sampling observation trajectories with video diffusion can lead to samples that are inconsistent with the overall joint distribution in Eqn (1). The observation trajectory $\tau_x^i$ not only affects the marginal likelihood of visual planning, but also the downstream likelihood of the action planning model.

**Consistency with Action Planning** To ensure observation trajectories $\tau_x^i$ that correctly maximize the joint distribution in Eqn (1), we optimize an observation trajectory that maximizes the following joint likelihood

$$(\tau_x^i)^* = \arg\max_{\tau_x^i} p_\phi(\tau_x^i|w_i, x_{i,1}) \prod_{t=1}^{T-1} p_\psi(a_{i,t}|x_{i,t}, x_{i,t+1}), \tag{4}$$

i.e. an image sequence that maximizes both conditional observation trajectory likelihood from video diffusion and the likelihood of sampled actions $\tau_a^i$ given the observation trajectory $\tau_x^i$.

To sample such an observation trajectory, we could iteratively bias the denoising of video diffusion using the log-likelihood of the sampled actions $\prod_{t=1}^{T-1} \log p_\psi(a_{i,t}|x_{i,t}, x_{i,t+1})$. While this solution is principled, it is slow as it requires sampling of entire action trajectories and calculating the corresponding likelihoods during every step of the denoising process. Thus, we approximate the sampling and the likelihood calculation of action trajectory $\prod_{t=1}^{T-1} p_\psi(a_{i,t}|x_{i,t}, x_{i,t+1})$ with a binary classifier $g_\psi(\tau_x^i)$ that models if the observation trajectory $\tau_x^i$ leads to a high-likelihood action trajectory.

We learn a binary classifier $g_\psi$ to assign high likelihood to feasible trajectories sampled from our video dataset $\tau_x^i \sim \mathcal{D}_{\text{video}}$ and low likelihood to infeasible trajectories generated by randomly shuffling the order of consecutive frames in feasible trajectories. Once trained, we can use the likelihood $\log g_\psi(1|\tau_x^i)$ to bias the denoising of the video diffusion and maximize the likelihood of the ensuing action trajectory. For further details on binary classifier, please refer to Appendix C.2.

### 2.3 Action Planning with Inverse Dynamics

After generating an observation trajectory $\tau_x^i$ from visual planning, our action planner generates an action trajectory $\tau_a^i$ from the observation trajectory. We leverage egocentric internet images for providing our action planner with useful visual priors. Our action planner is parameterized as an inverse dynamics model [1, 38] that infers the action $a_{i,t}$ given the observation pair $(x_{i,t}, x_{i,t+1})$:

$$a_{i,t} \sim p_\psi(a_{i,t}|x_{i,t}, x_{i,t+1})$$

**Training** To imbue the inverse dynamics $p_\psi$ with useful visual priors, we initialize it with VC-1 [32] weights, pretrained on ego-centric images and ImageNet. We then finetune it on dataset $\mathcal{D}_{\text{inv}} := \{\tau_x^i, \tau_a^i\}$ consisting of paired observation and action trajectories by optimizing:

$$\max_\psi \mathbb{E}_{\tau \in \mathcal{D}_{\text{inv}}} \left[ \log p_\psi(a_{i,t}|x_{i,t}, x_{i,t+1}) \right]$$

For further architectural details, please refer to Appendix B.3.

## 3 Experimental Evaluations

We evaluate the ability of `HiP` to solve long-horizon planning tasks that are drawn from distributions with substantial variation, including the number and types of objects and their arrangements. We then study the effects of iterative refinement and of pretraining on overall performance of `HiP`. We also compare against an alternative strategy of visually grounding the LLM without any task-specific data. In addition, we study how granularity of subgoals affects `HiP`'s performance, ablate over choice of visual planning model and analyze sensitivity of iterative refinement to hyperparameters (Appendix E).

### 3.1 Evaluation Environments

We evaluate `HiP` on three environments, `paint-block`, `object-arrange`, and `kitchen-tasks` which are inspired by combinatorial planning tasks in Mao et al. [33], Shridhar et al. [43] and Xing et al. [50] respectively.

- `paint-block`: A robot has to manipulate blocks in the environment to satisfy language goal instructions, such as *stack pink block on yellow block and place green block right of them*. However, objects of correct colors may not be present in the environment, in which case, the robot needs to first pick up white blocks and put them in the appropriate color bowls to paint them. After that, it should perform appropriate pick-and-place operations to stack a pink block on the yellow block and place the green block right of them. A new task $\mathcal{T}$ is generated by randomly selecting 3 final colors (out of 10 possible colors) for the blocks and then sampling a relation (out of 3 possible relations) for each pair of blocks. The precise locations of individual blocks, bowls, and boxes are fully randomized across different tasks. Tasks have $4 \sim 6$ subgoals.

- `object-arrange`: A robot has to place appropriate objects in the brown box to satisfy language goal instructions such as *place shoe, tablet, alarm clock, and scissor in brown box*. However, the environment may have distractor objects. Furthermore, some objects can be dirty, indicated by a lack of texture and yellow color. For these objects, the robot must first place them in a blue cleaning box and only afterwards place those objects in the brown box. A new task $\mathcal{T}$ is generated

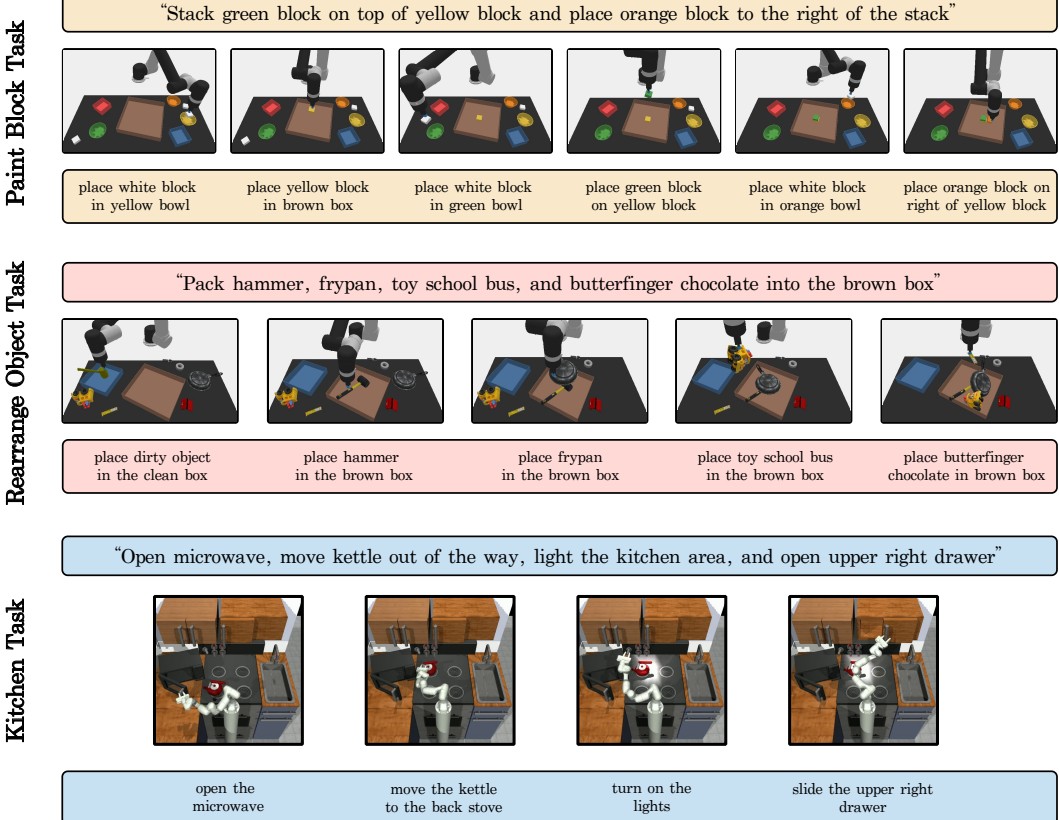

Figure 3: **Example Executions.** Example long-horizon generated plans on tasks in `paint-block`, `object-arrange`, and `kitchen-tasks` domains.

by randomly selecting 7 objects (out of 55 possible objects), out of which 3 are distractors, and then randomly making one non-distractor object dirty. The precise locations of individual objects and boxes are fully randomized across different tasks. Tasks usually have $3 \sim 5$ subgoals.

- `kitchen-tasks`: A robot has to complete kitchen subtasks to satisfy language goal instructions such as *open microwave, move kettle out of the way, light the kitchen area, and open upper right drawer*. However, the environment may have objects irrelevant to the subtasks that the robot must ignore. Furthermore, some kitchen subtasks specified in the language goal might already be completed, and the robot should ignore those tasks. There are 7 possible kitchen subtasks: opening the microwave, moving the kettle, switching on lights, turning on the bottom knob, turning on the top knob, opening the left drawer, and opening the right drawer. A new task $\mathcal{T}$ is generated by randomly selecting 4 out of 7 possible kitchen subtasks, randomly selecting an instance of microwave out of 3 possible instances, randomly selecting an instance of kettle out of 4 possible instances, randomly and independently selecting texture of counter, floor and drawer out of 3 possible textures and randomizing initial pose of kettle and microwave. With 50% probability, one of 4 selected kitchen subtask is completed before the start of the task. Hence, tasks usually have $3 \sim 4$ subtasks (i.e. subgoals).

**Train and Test Tasks** For all environments, we sample two sets of tasks $\mathcal{T}_{train}, \mathcal{T}_{test} \sim p(\mathcal{T})$. We use the train set of tasks $\mathcal{T}_{train}$ to create datasets $\mathcal{D}_{\text{classify}}, \mathcal{D}_{\text{video}}, \mathcal{D}_{\text{inv}}$ and other datasets required for training baselines. We ensure the test set of tasks $\mathcal{T}_{test}$ contains novel combinations of object colors in `paint-block`, novel combinations of object categories in `object-arrange`, and novel combinations of kitchen subtasks in `kitchen-tasks`.

**Evaluation Metrics** We quantitatively evaluate a model by measuring its task completion rate for `paint-block` and `object-arrange`, and subtask completion rate for `kitchen tasks`. We use the simulator to determine if the goal, corresponding to a task, has been achieved. We evaluate a model on $\mathcal{T}_{train}$ (seen) to test its ability to solve long-horizon tasks and on $\mathcal{T}_{test}$ (unseen) to test its ability to generalize to long-horizon tasks consisting of novel combinations of object colors in `paint-block`,

| Model | Paint-block | | Object-arrange | | Kitchen-tasks | |
|---|---|---|---|---|---|---|
| | **Seen** | **Unseen** | **Seen** | **Unseen** | **Seen** | **Unseen** |
| Transformer BC (oracle subgoals) | $8.3 \pm 1.9$ | $5.1 \pm 1.6$ | $10.2 \pm 2.9$ | $7.3 \pm 1.7$ | $48.4 \pm 21.6$ | $32.1 \pm 24.2$ |
| Gato (oracle subgoals) | $31.2 \pm 2.4$ | $28.6 \pm 2.9$ | $37.9 \pm 3.3$ | $36.5 \pm 3.2$ | $70.2 \pm 10.8$ | $66.8 \pm 12.2$ |
| Trajectory Transformer (oracle subgoals) | $22.1 \pm 2.1$ | $22.3 \pm 2.5$ | $30.5 \pm 2.3$ | $29.8 \pm 2.9$ | $66.4 \pm 20.7$ | $52.1 \pm 22.3$ |
| Action Diffuser (oracle subgoals) | $21.6 \pm 2.6$ | $18.2 \pm 2.3$ | $29.2 \pm 2.4$ | $27.6 \pm 2.1$ | $65.9 \pm 23.2$ | $55.1 \pm 22.8$ |
| HiP (Ours, oracle subgoals) | $\mathbf{81.2 \pm 1.8}$ | $\mathbf{79.6 \pm 1.9}$ | $\mathbf{91.8 \pm 2.9}$ | $\mathbf{92.3 \pm 2.3}$ | $\mathbf{92.8 \pm 7.1}$ | $\mathbf{89.8 \pm 7.6}$ |
| UniPi | $37.2 \pm 3.8$ | $35.3 \pm 3.2$ | $44.1 \pm 3.1$ | $44.2 \pm 2.9$ | $74.6 \pm 14.8$ | $73.4 \pm 11.2$ |
| SayCan | $67.2 \pm 3.3$ | $62.8 \pm 3.7$ | $70.3 \pm 2.6$ | $66.9 \pm 2.8$ | - | - |
| HiP (Ours) | $\mathbf{74.3 \pm 1.9}$ | $\mathbf{72.8 \pm 1.7}$ | $\mathbf{75 \pm 2.8}$ | $\mathbf{75.4 \pm 2.6}$ | $\mathbf{85.8 \pm 9.4}$ | $\mathbf{83.5 \pm 10.2}$ |

Table 1: **Performance on Long-Horizon tasks.** HiP not only outperforms the baselines in solving seen long-horizon tasks but its performance remains intact when solving unseen long-horizon tasks containing novel combination of objects colors in `paint-block`, novel combination of objects categories in `object-rearrange` and novel combination of subtasks in `kitchen-tasks`.

object categories in `object-arrange`, and kitchen subtasks in `kitchen-tasks`. We sample 1000 tasks from $\mathcal{T}_{train}$ and $\mathcal{T}_{test}$ respectively, and obtain average task completion rate on `paint-block` and `object-arrange` domains and average subtask completion rate on `kitchen tasks` domain. We report the mean and the standard error over 4 seeds in Table 1.

### 3.2 Baselines

There are several existing strategies for constructing robot manipulation policies conditioned on language goals, which we use as baselines in our experiments:

- **Goal-Conditioned Policy** A goal-conditioned transformer $a_{i,t} \sim p(a_{i,t}|x_{i,t}, w_i)$ that outputs action $a_{i,t}$ given a language subgoal $w_i$ and current observation $x_{i,t}$ (Transformer BC) [6]. We provide the model with oracle subgoals and encode these subgoals with a pretrained language encoder (Flan-T5-Base). We also compare against goal-conditioned policy with Gato [39] transformer.
- **Video Planner** A video diffusion model (UniPi) [12] $\{\tau_x^i\} \sim p(\{\tau_x^i\}|g, x_{i,1}), a_{i,t} \sim p(a_{i,t}|x_{i,t}, x_{i,t+1})$ that bypasses task planning, generates video plans for the entire task $\{\tau_x^i\}$, and infers actions $a_{i,t}$ using an inverse model.
- **Action Planners** Transformer models (Trajectory Transformer) [23] and diffusion models (Diffuser) [24, 4] $\{a_{i,t:T-1}\} \sim p(\{a_{i,t:T-1}\}|x_{i,t}, w_i)$ that produce an action sequence $\{a_{i,t:T-1}\}$ given a language subgoal $w_i$ and current visual observation $x_{i,t}$. We again provide the agents with oracle subgoals and encode these subgoals with a pretrained language encoder (Flan-T5-Base).
- **LLM as Skill Manager** A hierarchical system (SayCan) [2] with LLM as high level policy that sequences skills sampled from a repetoire of skills to accomplish a long-horizon task. We use CLIPort [43] policies as skills and the unnormalized logits over the pixel space it produces as affordances. These affordances grounds the LLM to current observation for producing next subgoal.

### 3.3 Results

We begin by comparing the performance of HiP and baselines to solve long-horizon tasks in `paint-block`, `object-arrange`, `kitchen-tasks` environments. Table 1 shows that HiP significantly outperforms the baselines, although the baselines have an advantage and have access to oracle subgoals. HiP's superior performance shows the importance of (i) hierarchy given it outperforms goal-conditioned policy (Transformer BC and Gato), (ii) task planning since it outperforms video planners (UniPi), and (iii) visual planning given it outperforms action planners (Trajectory Transformer, Action Diffuser). It also shows the importance of representing skills with video-based planners which can be pre-trained on Internet videos and can be applied to tasks (such as `kitchen-tasks`). SayCan, in contrast, requires tasks to be decomposed into primitives paired with an affordance function, which can be difficult to define for many tasks like the kitchen task. Thus, we couldn't run SayCan on `kitchen-tasks` environment. Finally, to quantitatively show how the errors in $f_\theta(x_{i,1}, w_i, g)$ affect the performance of HiP, we compare it to HiP with oracle subgoals. For further details on the training and evaluation of HiP, please refer to Appendix C. For implementation details on Gato and SayCan, please refer to Appendix D. For runtime analysis of HiP, please refer to Appendix F.

**Combinatorial Generalization to Unseen Long-horizon Tasks** We also quantitatively test the ability of HiP to generalize to unseen long-horizon tasks, consisting of novel combinations of object colors in `paint-block`, object categories in `object-arrange`, and subtasks in `kitchen-tasks`. Table 1 shows that HiP's performance remains intact when solving unseen long-horizon tasks, and

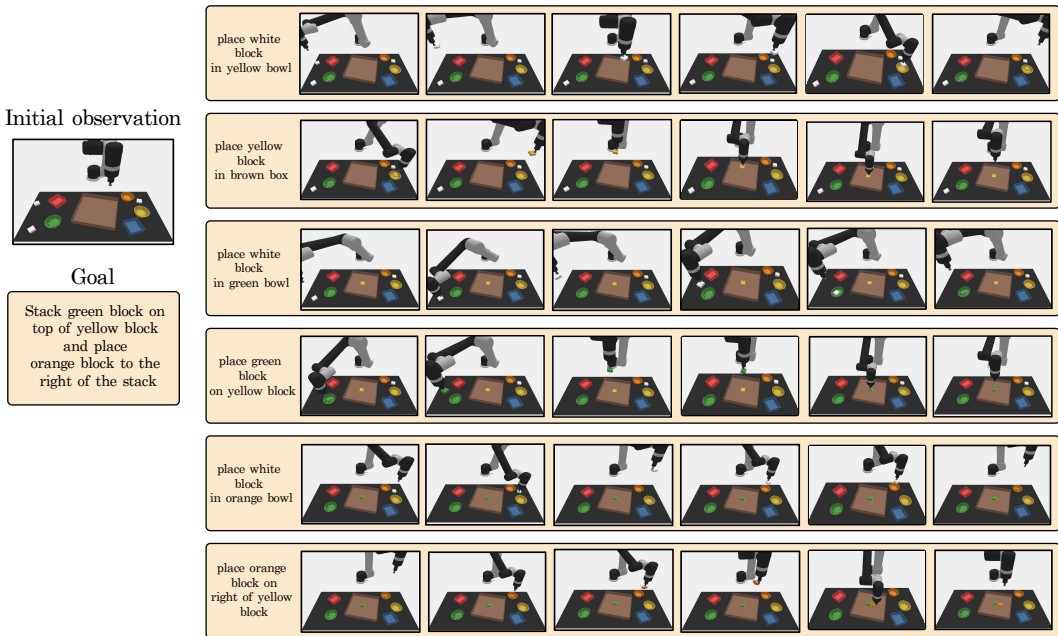

Figure 4: **Execution trajectory** of `HiP` on an novel long-horizon task in `paint-block` environment.

still significantly outperforms the baselines. Figure 4 visualizes the execution of `HiP` in unseen long-horizon tasks in `paint-block`.

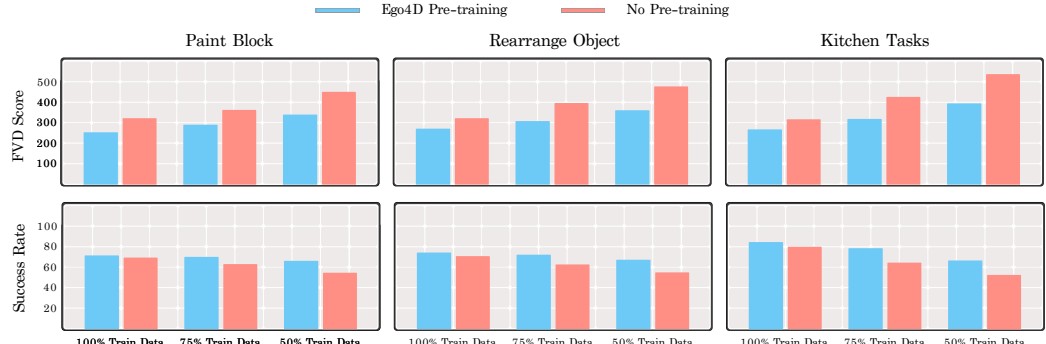

Figure 5: **Pretraining video diffusion model** with the Ego4D dataset consistently yields higher success rate and lower FVD scores (lower is better), even with reduced training dataset sizes. With pretraining, the model's FVD score escalates less gradually and its success rate falls less steeply as the dataset size shrinks.

**Pre-training Video Diffusion Model**   We investigate how much our video diffusion model benefits from pre-training on the Internet-scale data. We report both the success rate of `HiP` and Fréchet Video Distance (FVD) score that quantifies the similarity between generated videos and ground truth videos, where lower scores indicate greater similarity in Figure 5. We see that pretraining video diffusion leads to a higher success rate and lower FVD score. If we reduce the training dataset to $75\%$ and $50\%$ of the original dataset, the FVD score for video diffusion models (both, with and without Ego4D dataset pretraining) increases and their success rate falls. However, the video diffusion model with Ego4D dataset pretraining consistently gets higher success rate and lower FVD scores across different dataset sizes. As we decrease the domain-specific training data, it is evident that the gap in performance between the model with and without the Ego4D pre-training widens. For details on how we process the Ego4D dataset, please refer to Appendix C.2.

**Pre-training Inverse Dynamics Model**   We also analyze the benefit of pre-training our inverse dynamics model and report the mean squared error between the predicted and ground truth actions in Figure 6. The pre-training comes in the form of initializing the inverse dynamics model with

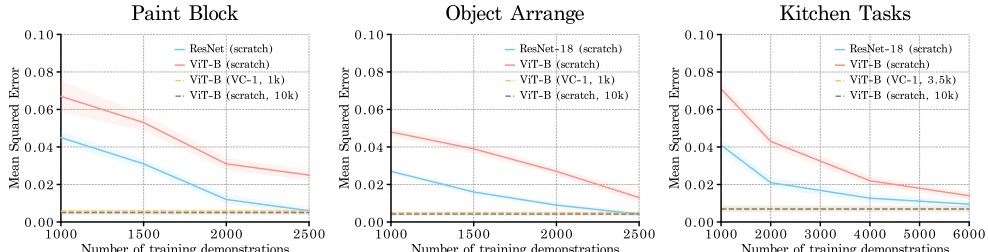

Figure 6: **Pretraining inverse dynamics model.** In `paint-block` and `object-arrange` (`kitchen-tasks`), when initialized with VC-1 weights, inverse dynamics model matches the performance of a randomly initialized model trained on 10K trajectories with just 1K (3.5k) trajectories. A smaller ResNet-18 model requires 2.5K (6k) trajectories to approach the same performance. The yellow and brown lines are overlaid on top of each other.

weights from VC-1 [32], a vision-transformer (ViT-B) [9] trained on ego-centric images with masked-autoencoding objective [16]. In `paint-block` and `object-arrange`, we see that the inverse dynamics, when initialized with weights from VC-1, only requires 1K labeled robotic trajectories to achieve the same performance as the inverse dynamics model trained on 10K labeled robotic trajectories but without VC-1 initialization. We also compare against an inverse dynamics model parameterized with a smaller network (ResNet-18). However, the resulting inverse dynamics model still requires 2.5K robotic trajectories to get close to the performance of the inverse dynamics model with VC-1 initialization in `paint-block` and `object-arrange`. In `kitchen-tasks`, inverse dynamics, when initialized with weights from VC-1, only requires 3.5k labeled robotic trajectories to achieve the same performance as the inverse dynamics model trained on 10K labeled robotic trajectories but without VC-1 initialization. When parameterized with ResNet-18, the inverse dynamics model still requires 6k robotic trajectories to get close to the performance of the inverse dynamics model with VC-1 initialization.

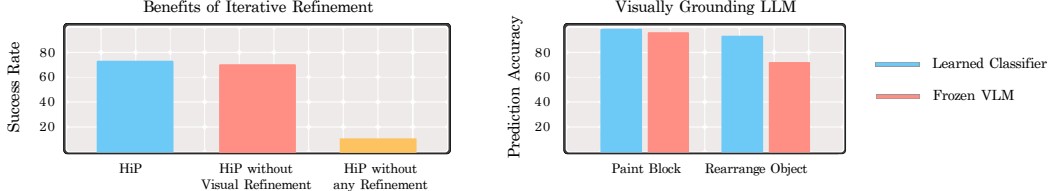

Figure 7: **Ablation Studies.** (Left) While task plan refinement is critical to `HiP`'s performance, visual plan refinement improves `HiP`'s performance by a smaller margin in `paint block` environment. (Right) While frozen pretrained VLM (MiniGPT4) matches the performance of a learned classifier in `paint-block` environment, its performance deteriorates in a more visually complex `object-arrange` environment.

**Importance of Task Plan and Visual Plan Refinements** We study the importance of refinement in task and visual planning in Figure 7. We compare to `HiP` without visual plan refinement and `HiP` without visual and task plan refinement in `paint block` environment. We see that task plan refinement for visual grounding of LLM is critical to the performance of `HiP`. Without it, the task plan is agnostic to the robot's observation and predicts subgoals that lead to erroneous visual and action planning. Furthermore, visual plan refinement improves the performance of `HiP` as well, albeit by a small margin. For a description of the hyperparameters used, please refer to Appendix C.4.

**Exploring Alternate Strategies for Visually Grounding LLM** We use a learned classifier $f_\theta(x_{i,1}, w_i, g)$ to visually ground the LLM. We explore if we can use a frozen pretrained Vision-Language Model (MiniGPT-4 [58]) as a classifier in place of the learned classifier. Although we didn't use any training data, we found the prompt engineering using the domain knowledge of the task to be essential in using the Vision-Language Model (VLM) as a classifier (see Appendix C.1 for details). We use subgoal prediction accuracy to quantify the performance of the learned classifier and the frozen VLM. Figure 7 illustrates that while both our learned multi-class classifier and frozen VLM perform comparably in the `paint-block` environment, the classifier significantly outperforms the VLM in the more visually complex `object-arrange` environment. We detail the two common failure modes of the VLM approach in `object-arrange` environment in Appendix C.1. As VLMs continue to improve, it is possible that their future versions match the performance of

learned classifiers and thus replace them in visually complex domains as well. For further details on the VLM parameterization, please refer to Appendix C.1.

# 4 Related Work

The field of foundation models for decision-making [52] has seen significant progress in recent years. A large body of work explored using large language models as zero-shot planners [21, 22, 29, 30, 2], but it is often difficult to directly ground the language model on vision. To address this problem of visually grounding the language model, other works have proposed to directly fine-tune large language models for embodied tasks [28, 10]. However, such an approach requires large paired vision and language datasets that are difficult to acquire. Most similar to our work, SayCan [2] uses an LLM to hierarchically execute different tasks by breaking language goals into a sequence of instructions, which are then inputted to skill-based value functions. While SayCan assumes this fixed set of skill-based value functions, our skills are represented as video-based planners [12], enabling generalization to new skills.

Another set of work has explored how to construct continuous space planners with diffusion models [25, 4, 49, 55, 48, 57, 12]. Existing works typically assume task-specific datasets from which the continuous-space planner is derived [24, 3, 55]. Most similar to our work, UniPi [12] proposes to use videos to plan in image space and similarly relies on internet videos to train image space planners. We build on top of UniPi to construct our foundation model for hierarchical planning, and illustrate how UniPi may be combined with LLMs to construct longer horizon continuous video plans.

Moreover, some works [28, 54, 5] explored how different foundation models may be integrated with each other. In Flamingo [5], models are combined through joint finetuning with paired datasets, which are difficult to collect. In contrast both Zeng et al. [54] and Li et al. [28] combine different models zero-shot using either language or iterative consensus. Our work proposes to combine language, video, and ego-centric action models together by taking the product of their learned distributions [11]. We use a similar iterative consensus procedure as in Li et al. [28] to sample from the entire joint distribution and use this combined distribution to construct a hierarchical planning system.

# 5 Limitations and Conclusion

**Limitations**   Our approach has several limitations. As high-quality foundation models for visual sequence prediction and robot action generation do not exist yet, our approach relies on smaller-scale models that we directly train. Once high-quality video foundation models are available, we can use them to guide our smaller-scale video models [51] which would reduce the data requirements of our smaller-scale video models. Furthermore, our method uses approximations to sample from the joint distribution between all the model. An interesting avenue for future work is to explore more efficient and accurate methods to ensure consistent samples from the joint distribution.

**Conclusion**   In this paper, we have presented an approach to combine many different foundation models into a consistent hierarchical system for solving long-horizon robotics problems. Currently, large pretrained models are readily available in the language domain only. Ideally, one would train a foundation model for videos and ego-centric actions, which we believe will be available in the near future. However, our paper focuses on leveraging separate foundation models trained on different modalities of internet data, instead of training a single big foundation model for decision making. Hence, for the purposes of this paper, given our computational resource limitations, we demonstrate our general strategy with smaller-scale video and ego-centric action models trained in simulation, which serve as proxies for larger pretrained models. We show the potential of this approach in solving three long-horizon robot manipulation problem domains. Across environments with novel compositions of states and goals, our method significantly outperforms the state-of-the-art approaches towards solving these tasks.

In addition to building larger, more general-purposed visual sequence and robot control models, our work suggests the possibility of further using other pretrained models in other modalities, such as touch and sound, which may be jointly combined and used by our sampling approach. Overall, our work paints a direction towards decision making by leveraging many different powerful pretrained models in combination with a tiny bit of training data. We believe that such a system will be substantially cheaper to train and will ultimately result in more capable and general-purpose decision making systems.

## Acknowledgements

The authors would like to thank the members of Improbable AI Lab for discussions and helpful feedback. We thank MIT Supercloud and the Lincoln Laboratory Supercomputing Center for providing compute resources. This research was supported by an NSF graduate fellowship, a DARPA Machine Common Sense grant, ARO MURI Grant Number W911NF-21-1-0328, ARO MURI W911NF2310277, ONR MURI Grant Number N00014-22-1-2740, and an MIT-IBM Watson AI Lab grant. The views and conclusions contained in this document are those of the authors and should not be interpreted as representing the official policies, either expressed or implied, of the United States Army Research Office, the Office of Naval Research, or the U.S. Government. The U.S. Government is authorized to reproduce and distribute reprints for Government purposes, notwithstanding any copyright notation herein.

## Author Contributions

**Anurag Ajay** co-conceived the framework of leveraging pretrained foundation models for decision making, implemented visual planning and action planning in HiP, evaluated HiP on long-horizon tasks, performed ablation studies and helped in paper writing.

**Seungwook Han** co-conceived in conceiving the framework of leveraging pretrained foundation models for decision making, implemented task planning in HiP and helped in paper writing.

**Yilun Du** co-conceived the framework of leveraging pretrained foundation models for decision making, implemented trajectory transformer and transformer BC and evaluated them on long-horizon tasks, implemented data generation scripts for object arrange and paint block environment, and lead paper writing.

**Shuang Li** helped in conceiving the idea of iterative refinement for consistency between pretrained foundation models, participated in research discussions and helped in writing paper.

**Abhi Gupta** participated in research discussions and helped in making figures.

**Tommi Jaakkola** participated in research discussions.

**Joshua Tenenbaum** participated in research discussions.

**Leslie Kaelbling** participated in research discussions, suggested baselines and ablation studies, conceived the structure of the paper and helped in paper writing.

**Akash Srivastava** participated in research discussions, suggested the idea of using classifier for consistency between observations and the large language model and provided feedback on paper writing.

**Pulkit Agrawal** was involved in research discussions, suggested ablation studies related to iterative refinement, provided feedback on writing, positioning of the work and overall advising.

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

# Appendix

In this Appendix, we discuss how we factorize the hierarchical decision-making process in Section A. In Section B, we then detail the background and architecture for visually grounded task planning, visual planning with video diffusion, and action planning with inverse dynamics model. In Section C, we discuss the training and evaluation details for the different levels of planning in `HiP` and the corresponding training hyperparameters. In Section D, we discuss implementation details for Gato and SayCan. In Section E, we showcase additional ablation studies comparing different approaches to enforce consistency across the levels of hierarchy, analyzing effect of granularity of subgoals on performance of `HiP`, ablating on the choice of video planning model and analyzing sensitivity of iterative refinement to hyperparameters. Finally, in Section F, we analyze runtime of different components of `HiP`.

## A  Factorizing Hierarchical Decision-Making Process

We model the hierarchical decision-making process described in Section 2 with $p_\Theta$ which can be factorized into the task distribution $p_\theta$, visual distribution $p_\phi$, and action distribution $p_\psi$.

$$p_\Theta(W, \{\tau_x^i\}, \{\tau_a^i\}|g, x_{1,1}) = \prod_{i=1}^{N} p_\theta(w_i|g, x_{i,1}, w_{<i}, \tau_x^{<i}, \tau_a^{<i}) \prod_{i=1}^{N} p_\phi(\tau_x^i|w_{\leq i}, x_{i,1}, g, \tau_x^{<i}, \tau_a^{<i})$$
$$\prod_{i=1}^{N} p_\psi(\tau_a^i|\tau_x^{\leq i}, w_{\leq i}, x_{i,1}, g, \tau_a^{<i})$$

Here, given random variables $Y^i$, $Y^{<i}$ and $Y^{\leq i}$ represents $\{Y^1, \ldots, Y^{i-1}\}$ and $\{Y^1, \ldots, Y^i\}$ respectively. Now, we apply Markov assumption: Given current observation $x_{i,1}$, future variables $(w_i, \tau_x^i, \tau_a^i)$ and past variables $(w_j, \tau_x^j, \tau_a^j \ \ \forall j < i)$ are conditionally independent.

$$p_\Theta(W, \{\tau_x^i\}, \{\tau_a^i\}|g, x_{1,1}) = \prod_{i=1}^{N} p_\theta(w_i|g, x_{i,1}) \prod_{i=1}^{N} p_\phi(\tau_x^i|w_i, x_{i,1}, g) \prod_{i=1}^{N} p_\psi(\tau_a^i|\tau_x^i, w_i, x_{i,1}, g)$$

We model task distribution $p_\theta$ with a large language model (LLM) which is independent of observation $x_{i,1}$. Since the image trajectory $\tau_x^i = \{x_{i,1:T}\}$ describes a physically plausible plan for achieving subgoal $w_i$ from observation $x_{i,1}$, it is conditionally independent of goal $g$ given subgoal $w_i$ and observation $x_{i,1}$. Furthermore, we assume that an action $a_{i,t}$ can be recovered from observation at the same timestep $x_{i,t}$ and the next timestep $x_{i,t+1}$. Thus, we can write the factorization as

$$p_\Theta(W, \{\tau_x^i\}, \{\tau_a^i\}|g, x_{1,1}) = \left(\prod_{i=1}^{N} p_\theta(w_i|g)\right) \left(\prod_{i=1}^{N} p_\phi(\tau_x^i|w_i, x_{i,1})\right) \left(\prod_{i=1}^{N} \prod_{t=1}^{T-1} p_\psi(a_{i,t}|x_{i,t}, x_{i,t+1})\right)$$

## B  Background and Architecture

### B.1  Task Planning

**Background on Density Ratio Estimation**   Let $p$ and $q$ be two densities, such that $q$ is absolutely continuous with respect to $p$, denoted as $q << p$ i.e. $q(\boldsymbol{x}) > 0$ wherever $p(\boldsymbol{x}) > 0$. Then, their ratio is defined as $r(\boldsymbol{x}) = p(\boldsymbol{x})/q(\boldsymbol{x})$ over the support of $p$. We can estimate this density ratio $r(\boldsymbol{x})$ by training a binary classifier to distinguish between samples from $p$ and $q$ [46, 14, 17]. More recent work [45] has shown one can introduce auxiliary densities $\{m_i\}_{i=1}^{M}$ and train a multi-class classifier to distinguish samples between $M$ classes to learn a better-calibrated and more accurate density ratio estimator. Once trained, the log density ratio can be estimated by $\log r(\boldsymbol{x}) = \hat{h_p}(\boldsymbol{x}) - \hat{h_q}(\boldsymbol{x})$, where $\hat{h_i}(\boldsymbol{x})$ is the unnormalized log probability of the input sample under the $i^{th}$ density, parameterized by the model.

**Learning a Classifier to Visually Ground Task Planning**   We estimate the density ratio $\frac{p(x_{i,1}|w_i, g)}{p(x_{i,1}|g)}$ with a multi-class classifier $f_\phi(x_{i,1}, \{w_j\}, g)$ trained to distinguish samples amongst the conditional distributions $p(x_{i,1}|w_i, g), \ldots, p(x_{i,1}|w_M, g)$ and the marginal distribution $p(x_{i,1}|g)$. Upon convergence, the classifier learns to assign high scores to $(x_{i,1}, w_i, g)$ if $w_i$ is the subgoal corresponding to the observation $x_{i,1}$ and task $g$ and low scores otherwise.

**Architecture** We parameterize $f_\phi$ as a 4-layer multi-layer perceptron (MLP) on top of an ImageNet-pretrained vision encoder (ResNet-18 [15]) and a frozen pretrained language encoder (Flan-T5-Base [8]). The vision encoder encodes the observation $x_{i,1}$, and the text encoder encodes the subgoals $w_j$ and the goal $g$. The encoded observation, the encoded subgoals, and the encoded goal are concatenated, and passed through a MLP with 3 hidden layers of sizes 512, 256, and 128. The output dimension for MLP (i.e., number of classes for multi-classification) $M$ is 6 for `paint-block` environment, 5 for `object-arrange` environment and 4 for `kitchen-tasks` environment.

**Choice of Large Language Model** We use GPT3.5-turbo [37] as our large language model.

## B.2 Visual Planning

**Background** Diffusion Probabilistic Models [44, 19] learn the data distribution $h(\boldsymbol{x})$ from a dataset $\mathcal{D} := \{\boldsymbol{x}^i\}$. The data-generating procedure involves a predefined forward noising process $q(\boldsymbol{x}_{k+1}|\boldsymbol{x}_k)$ and a trainable reverse process $p_\phi(\boldsymbol{x}_{k-1}|\boldsymbol{x}_k)$, both parameterized as conditional Gaussian distributions. Here, $\boldsymbol{x}_0 := \boldsymbol{x}$ is a sample, $\boldsymbol{x}_1, \boldsymbol{x}_2, ..., \boldsymbol{x}_{K-1}$ are the latents, and $\boldsymbol{x}_K \sim \mathcal{N}(\boldsymbol{0}, \boldsymbol{I})$ for a sufficiently large $K$. Starting with Gaussian noise, samples are then iteratively generated through a series of "denoising" steps. Although a tractable variational lower-bound on $\log p_\phi$ can be optimized to train diffusion models, Ho et al. [19] propose a simplified surrogate loss:

$$\mathcal{L}_{\text{denoise}}(\theta) := \mathbb{E}_{k \sim [1,K], \boldsymbol{x}_0 \sim h, \epsilon \sim \mathcal{N}(\boldsymbol{0}, \boldsymbol{I})}[||\epsilon - \epsilon_\phi(\boldsymbol{x}_k, k)||^2]$$

The predicted noise $\epsilon_\theta(\boldsymbol{x}_k, k)$, parameterized with a deep neural network, estimates the noise $\epsilon \sim \mathcal{N}(0, I)$ added to the dataset sample $\boldsymbol{x}_0$ to produce noisy $\boldsymbol{x}_k$.

**Guiding Diffusion Models with Text** Diffusion models are most notable for synthesizing high-quality images [41, 35] and videos [20, 53] from text descriptions. Modeling the conditional data distribution $q(\boldsymbol{x}|\boldsymbol{y})$ makes it possible to generate samples satisfying the text description $\boldsymbol{y}$. To enable conditional data generation with diffusion, Ho and Salimans [18] modified the original training setup to learn both a conditional $\epsilon_\phi(\boldsymbol{x}_k, \boldsymbol{y}, k)$ and an unconditional $\epsilon_\phi(\boldsymbol{x}_k, k)$ model for the noise. The unconditional noise is represented, in practice, as the conditional noise $\epsilon_\phi(\boldsymbol{x}_k, \emptyset, k)$, where a dummy value $\emptyset$ takes the place of $\boldsymbol{y}$. The perturbed noise $\epsilon_\phi(\boldsymbol{x}_k, \emptyset, k) + \omega(\epsilon_\phi(\boldsymbol{x}_k, \boldsymbol{y}, k) - \epsilon_\phi(\boldsymbol{x}_k, \emptyset, k))$ (i.e. *classifier-free guidance*) is used to later generate samples.

**Video Diffusion in Latent Space** As diffusion models generally perform denoising in the input space [19], the optimization and inference become computationally demanding when dealing with high-dimensional data, such as videos. Inspired by recent works [40, 53], we first use an autoencoder $v_{\text{enc}}$ to learn a latent space for our video data. It projects an observation trajectory $\tau_x$ (i.e., video) into a 2D tri-plane representation [53] $\tau_z = [\tau_z^T, \tau_z^H, \tau_z^W]$ where $\tau_z^T, \tau_z^H, \tau_z^W$ capture variations in the video across time, height, and width respectively. We then diffuse over this learned latent space [53].

**Latent Space Video Diffusion for Visual Planning** Our video diffusion model $p_\phi(\tau_x^i|w_i, x_{i,1})$ generates video $\tau_x^i$ given a language subgoal $w_i$ and the current observation $x_{i,1}$. It is parameterized through its noise model $\epsilon_\phi((\tau_z^i)_k, w_i, x_{i,1}, k) := \epsilon_\phi((\tau_z^i)_k, l_{\text{enc}}(w_i), v_{\text{enc}}(x_{i,1}), k)$ where $\tau_z^i := v_{\text{enc}}(\tau_x^i)$ is the latent representation of video $\tau_x^i$ over which we diffuse. We condition the noise model $\epsilon_\phi$ on subgoal $w_i$ using a pretrained language encoder $l_{\text{enc}}$ and on current observation $x_{i,1}$ using video encoder $v_{\text{enc}}$. To use $v_{\text{enc}}$ with a single observation $x_{i,1}$, we first tile the observation along the temporal dimension to create a video.

**Architecture** We now detail the architectures of different components:

- **Video Autoencoder** We borrow our architecture for $v_{\text{enc}}$ from PVDM [53] which uses transformers to project video $\tau_x \in \mathbb{R}^{T \times H \times W}$ to latent codes $\tau_z = [\tau_z^T, \tau_z^H, \tau_z^W]$ where $\tau_z^T \in \mathbb{R}^{C \times H' \times W'}$, $\tau_z^H \in \mathbb{R}^{C \times T \times W'}$, $\tau_z^W \in \mathbb{R}^{C \times H' \times T}$. Here, $T = 50$ represents the time horizon of a video, $H = 48$ represents video height, $W = 64$ represents video width, $C = 4$ represents latent codebook dimension, $H' = 12$ represents latent height, and $W' = 8$ represents latent width.

- **Language Encoder** We use Flan-T5-Base [8] as the pretrained frozen language encoder $l_{\text{enc}}$.

- **Noise Model** We borrow PVDM-L architecture [53] which uses 2D UNet architecture, similar to the one in Latent Diffusion Model (LDM) [40], to represent $p(\tau_z|\tau_z')$. In our case, $\tau_z = v_{\text{enc}}(\tau_x^i)$ and $\tau_z' = v_{\text{enc}}(x_{i,1})$. To further condition noise model $\epsilon_\phi$ on $l_{\text{enc}}(w_i)$, we augment the 2D UNet Model with cross-attention mechanism borrowed by LDM [40].

For implementing these architectures, we used the codebase https://github.com/sihyun-yu/PVDM which contains the code for PVDM and LDM.

**Classifier for Consistency between Visual Planning and Action Planning**   To ensure consistency between visual planning and action planning, we want to sample observation trajectories that maximizes both conditional observation trajectory likelihood from diffusion and the likelihood of sampled actions given the observation trajectory (see equation 4). To approximate likelihood calculation of action trajectory, we learn a binary classifier $g_\psi$ that models if the observation trajectory leads to a high likelihood action trajectories. Since diffusion happens in latent space and we use gradients from $g_\psi$ to bias the denoising of the video diffusion, $g_\psi(\tau_z^i)$ takes the observation trajectory in latent space. The binary classifier $g_\psi$ is trained to distinguish between observation trajectories in latent space sampled from video dataset $\tau_z^i = v_{\text{enc}}(\tau_x^i), \tau_x^i \sim \mathcal{D}_{\text{video}}$ (i.e. label of 1) and observation trajectories in latent space sampled from video dataset whose frames where randomly shuffled $(\tau_z^i)' = v_{\text{enc}}(\sigma(\tau_x^i)), \tau_x^i \sim \mathcal{D}_{\text{video}}$ (i.e. label of 0). Here, $\sigma$ denotes the random shuffling of frames. To randomly shuffle frames in an observation trajectory (of length 50), we first randomly select 5 frames in the observation trajectory. For each of the selected frame, we randomly permute it with its neighboring frame (i.e. either with the frame before it or with the frame after it). Once $g_\psi$ is trained, we use it to bias the denoising of the video diffusion

$$\hat{\epsilon} := \epsilon_\phi((\tau_z)_k, v_{\text{enc}}(x_t), k) + \omega(\epsilon_\phi((\tau_z)_k, v_{\text{enc}}(x_t), l_{\text{enc}}(w), k) - \epsilon_\phi((\tau_z)_k, v_{\text{enc}}(x_t), k))$$
$$- \omega' \nabla_{(\tau_z)_k} \log g_\psi(1|(\tau_z)_k)$$

Here, $\hat{\epsilon}$ is the noise used in denoising of the video diffusion and $\omega, \omega'$ are guidance hyperparameters.

**Classifier Architecture**   The classifier $g_\psi(\tau_z = [\tau_z^T, \tau_z^H, \tau_z^W])$ has a ResNet-9 encoder that converts $\tau_z^T, \tau_z^H$, and $\tau_z^W$ to latent vectors, then concatenate those latent vectors and passes the concatenated vector through an MLP with 2 hidden layers of sizes 256 and 128 and an output layer of size 1.

### B.3   Action Planning

To do action planning, we learn an inverse dynamics model to $p_\psi(a_{i,t}|x_{i,t}, x_{i,t+1})$ predicts 7-dimensional robot states $s_{i,t} = p_\psi(x_{i,t})$ and $s_{i,t+1} = p_\psi(x_{i,t+1})$. The first 6 dimensions of the robot state represent joint angles and the last dimension of the robot state represents the gripper state (i.e., whether it's open or closed). The first 6 action dimension is represented as joint angle difference $a_{i,t}[: 6] = s_{i,t+1}[: 6] - s_{i,t}[: 6]$ while the last action dimension is gripper state of next timestep $a_{i,t}[-1] = s_{i,t+1}[-1]$.

**Architecture**   We use ViT-B [9] (VC-1 [32] initialization) along with a linear layer to parameterize $p_\psi$. ViT-B projects the observation $x_{i,t} \in \mathbb{R}^{48 \times 64 \times 3}$ to 768 dimensional latent vector from which the linear layer predicts the 7 dimensional state $s_{i,t}$.

## C   Training and Evaluation

### C.1   Task Planning

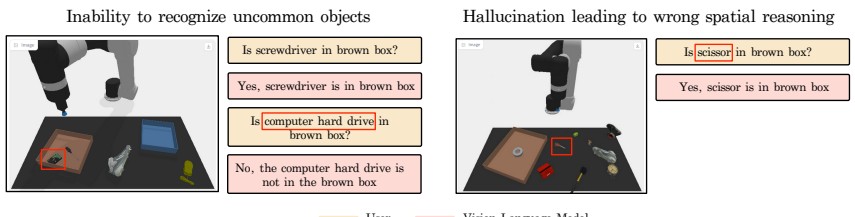

Figure 8: **Failure in VLM.** to recognize uncommon objects like computer hard drives and occasional hallucination of object presence, leading to incorrect visual reasoning.

**Training Objective and Dataset for Learned Classifier**   We use a softmax cross-entropy loss to train the multi-class classifier $f_\phi(x_{i,1}, \{w_j\}_{j=1}^M, g)$ to classify an observation $x_{i,1}$ into one of the $M$ given subgoal. We train it using the classification dataset $\mathcal{D}_{\text{classify}} := \{x_{i,1}, g, \{w_j\}_{j=1}^M, i\}$ consisting of observation $x_{i,1}$, goal $g$, candidate subgoals $\{w_j\}_{j=1}^M$ and the correct subgoal label $i$. The classification dataset for `paint-block`, `object-arrange`, and `kitchen-tasks` consists of 58k, 82k and 50k datapoints respectively.

**Vision-Language Model (VLM) as a Classifier**   We use a frozen pretrained Vision-Language Model (VLM) (MiniGPT4 [58]) as a classifier. We first sample a list of all possible subgoals

$W = \{w_i\}_{i=1}^M$ from the LLM given the language goal $g$. We then use the VLM to eliminate subgoals from $W$ that have been completed. For each subgoal, we question the VLM whether that subgoal has been completed. For example, consider the subgoal "Place white block in yellow bowl". To see if the subgoal has been completed, we ask the VLM "Is there a block in yellow bowl?". Consider the subgoal "Place green block in brown box" as another example. To see if the subgoal has been completed, we ask the VLM "Is there a green block in brown box?". Furthermore, if the VLM says "yes" and the subgoal has been completed, we also remove other subgoals from $W$ that should have been completed, such as "Place white block in green bowl". Once we have eliminated the completed subgoals, we use the domain knowledge to determine which subgoal to execute out of all the remaining subgoals. As an example, if the goal is to "Stack green block on top of blue block in brown box" and we have a green block in green bowl and a blue block in blue bowl, we should execute the subgoal "Place blue block in brown box" before the subgoal "Place green block on blue block". While this process of asking questions from VLM to determine the remaining subgoals and then sequencing the remaining subgoals doesn't require any training data, it heavily relies on the task's domain knowledge.

**Failure Modes of VLM**    We observe two common failure modes of the VLM approach in `object-arrange` environment and visualize them in Figure 8. First, because the model is not trained on any in-domain data, it often fails to recognize uncommon objects, such as computer hard drives, in the observations. Second, it occasionally hallucinates the presence of objects at certain locations and thus leads to incorrect visual reasoning.

**VLM as a Subgoal Predictor**    We also tried to prompt the VLM with 5 examples of goal $g$ and subgoal candidates $\{w_i\}_{i=1}^M$ and then directly use it to generate the next subgoal $w_i$ given the observation $x_{i,1}$ and the goal $g$. However, it completely failed. We hypothesize that the VLM fails to directly generate the next subgoal due to its inability to perform in-context learning.

**Evaluation**    We evaluate the trained classifier $f_\phi$ and the frozen VLM for subgoal prediction accuracy on 5k unseen datapoints, consisting of observation, goal, candidate subgoals and correct subgoal, generated from test tasks $\mathcal{T}_{\text{test}}$. We average over 4 seeds and show the results in Figure 7.

## C.2    Visual Planning

**Ego4D dataset processing**    We pre-train on *canonical clips* of the Ego4D dataset which are text-annotated short clips made from longer videos. We further divide each canonical clip into 10sec segments from which we derive 50 frames. We resize each frame to $48 \times 64$. We create a pretraining Ego4D dataset of (approximately) 344k short clips, each consisting of 50 frames and a text annotation. We use the loader from R3M [34] codebase (https://github.com/facebookresearch/r3m) to load our pretraining Ego4D dataset.

**Training Objective and Dataset**    We use pixel-level L1 reconstruction and negative perceptual similarity for training the autoencoder $v_{\text{enc}}$. We borrow this objective from PVDM [53] paper except we don't use adversarial loss. We keep the language encoder frozen. We use denoising loss in video latent space $\mathbb{E}_{k \sim [1,K], \tau_z, w, x \sim \mathcal{D}, \epsilon \sim \mathcal{N}(O,I)}[\|\epsilon - \epsilon_\phi((\tau_z)_k, l_{\text{enc}}(w), v_{\text{enc}}(x), k)\|^2]$ to train the noise model $\epsilon_\phi$. We replace $w$ with a null token so that $\epsilon_\phi$ learns both a text-conditional model and an unconditional model. We pretrain the autoencoder $v_{\text{enc}}$ and the noise model $\epsilon_\phi$ on the processed Ego4D dataset. We then finetune it on our dataset $\mathcal{D}_{\text{video}} := \{\tau_x^i, w_i\}$ consisting of approximately 100k observation trajectories of length $T = 50$ and associated text subgoals.

**Classifier Training Objective and Dataset**    We use a binary cross-entropy loss to train the binary classifier $g_\psi(\tau_z)$ that predicts if the observation trajectory in latent space $\tau_z = v_{\text{enc}}(\tau_x)$ leads to high-likelihood action trajectory. It is trained using trajectories from video dataset $\tau_x \sim \mathcal{D}_{\text{video}}$.

## C.3    Action Planning

**Training Objective and Dataset**    We train inverse dynamics $p_\psi$ on a dataset $\mathcal{D}_{\text{inv}}$. Since actions are differences between robotic joint states, we train $p_\psi$ to directly predict robotic state $s_{i,t} = p_\psi(x_{i,t})$ by minimizing the mean squared error between the predicted robotic state and ground truth robotic state. Hence, $\mathcal{D}_{\text{inv}} := \{\tau_x^i, \tau_s^i\}$ consists of 1k paired observation and robotic state trajectories, each having a length of $T = 50$, in `paint-block` and `object-arrange` domains. In `kitchen-tasks` domain, it consists of 3.5k paired observation and robotic state trajectories, each having a length of $T = 50$.

**Evaluation**     We evaluate the trained $p_\psi$ (i.e., VC-1 initialized model and other related models in Figure 6) on 100 unseen paired observation and robotic state trajectories generated from test tasks $\mathcal{T}_{\text{test}}$. We use mean squared error to evaluate our inverse dynamics models. We use 4 seeds to calculate the standard error, represented by the shaded area in Figure 6.

### C.4   Hyperparameters

**Task Planning**     We train $f_\phi$ for 50 epochs using AdamW optimizer [31], a batch size of 256, a learning rate of $1e - 3$ and a weight decay of $1e - 6$. We used one V100 Nvidia GPU for training the multi-class classifier.

**Visual Planning**     We borrow our hyperparameters for training video diffusion from the PVDM paper [53]. We use AdamW optimizer [31], a batch size of 24 and a learning rate of $1e - 4$ for training the autoencoder. We use AdamW optimizer, a batch size of 64, and a learning rate of $1e - 4$ for training the noise model. During the pretraining phase with the Ego4D dataset, we train the autoencoder for 5 epochs and then the noise model for 5 epochs. During the finetuning phase with $\mathcal{D}_{\text{video}}$, we train the autoencoder for 10 epochs and then the noise model for 40 epochs. We used two A6000 Nvidia GPUs for training these diffusion models. We train $g_\psi$ for 10 epochs using AdamW optimizer, a batch size of 256 and a learning rate of $1e - 4$. We used one V100 Nvidia GPU for training the binary classifier. During classifier-free guidance, we use $\omega = 4$ and $\omega' = 1$.

**Action Planning**     We train VC-1 initialized inverse dynamics model for 20 epochs with AdamW optimizer [31], a batch size of 256 and a learning rate of $3e - 5$. We trained other randomly initialized ViT-B inverse dynamics models and randomly initialized ResNet-18 inverse dynamics models for 20 epochs with AdamW optimizer, a batch size of 256, and a learning rate of $1e - 4$. We used one V100 Nvidia GPU for training these inverse dynamics models.

## D   Implementation Details for Gato and SayCan

In training the visual and action planning for `HiP`, we use 100k robot videos for visual planner and the inverse dynamics, when trained from scratch, utilizes 10k state-action trajectory pairs. In order to ensure fair comparison, we use 110k datapoints for training Gato [39] and SayCan [2].

**Gato**     We borrow the Gato [39] architecture from Vima Codebase and use it for training a language conditioned policy with imitation learning. We use 110k (langauge, observation trajectory, action trajectory) datapoints in each of the three domains for training Gato. Furthermore, we provide oracle subgoals to Gato.

**SayCan**     We borrow the SayCan [2] algorithm from the SayCan codebase and adapt it to our settings. Following the recommendations of SayCan codebase, we use CLIPort policies as primitives. CLIPort policies take in top-down RGBD view and outputs pick and place pixel coordinates. Then, an underlying motion planner picks the object from the specified pick-coordinate and places the object at the specified place-coordinate. We train CLIPort policies on 110k (language, observation, action) datapoints in `paint-block` and `object-arrange` domain. The SayCan paper uses value function as an affordance function to select the correct subgoal given current observation and high level goal. However, CLIPort policies don't have a value function. The SayCan codebase uses a hardcoded scoring function which doesn't apply to `object-arrange` domain. To overcome these issues, we use the LLM grounding strategy from **?** ]. It uses unnormalized logits over the pixel space given by CLIPort policies as affordance and uses it to ground LLM to current observation and thus predict the subgoal. We then compare SayCan with `HiP` and other baselines on `paint-block` and `object-arrange` domain in Table 1. While SayCan outpeforms other baselines, `HiP` still outperforms it both on seen and unseen tasks of `paint-block` and `object-arrange` domain. We couldn't run SayCan on `kitchen-tasks` domain as there's no clear-cut primitive in that domain. This points to a limitation of SayCan which requires tasks to be expressed in terms of primitives with each primitive paired with an affordance function.

## E   Additional Ablation Studies

### E.1   Consistency between task planning and visual planning

To make task planning consistent with visual planning, we need to select subgoal $w_i^*$ which maximizes the joint likelihood (see equation 2) of LLM $p_{\text{LLM}}(w_i|g)$ and video diffusion $p_\phi(\tau_x^i|w_i, x_{i,1})$. While

generating videos for different subgoal candidates $w_i$ and calculating the likelihood of the generated video is computationally expensive, we would still like to evaluate its performance in subgoal prediction given it is theoretically grounded. To this end, we first sample $M$ subgoals $W = \{w_j\}_{j=1}^M$ from the LLM. Then, we calculate $w_i^* = \arg\max_{w \in W} \log p_\phi(\tau_x^i|w, x_{i,1})$ and use $w_i^*$ as our predicted subgoal. Since $\log p_\phi(\tau_x^i|w, x_{i,1})$ is intractable, we estimate its variational lower-bound as an approximation. We use this approach for subgoal prediction in `paint-block` environment and compare its performance to that of the learned classifier. It achieves a subgoal prediction accuracy of $54.3 \pm 7.2\%$ whereas the learned classifier achieves a subgoal prediction accuracy of $98.2 \pm 1.5\%$ in `paint-block` environment. Both approaches outperform the approach of randomly selecting a subgoal from $W$ (i.e., no task plan refinement), which yields a subgoal prediction accuracy of $16.67\%$ given $M = 6$. The poor performance of the described approach could result from the fact that the diffusion model only coarsely approximates the true distribution $p(\tau_x^i|w_i, x_{i,1})$, which results in loose variational lower-bound and thus uncalibrated likelihoods from the diffusion model. A larger diffusion model could better approximate $p(\tau_x^i|w_i, x_{i,1})$, resulting in tighter variational lower-bound and better-calibrated likelihoods.

### E.2 Consistency between visual planning and action planning

To make visual planning consistent with action planning, we need to select observation trajectory $(\tau_x^i)^*$ which maximizes joint likelihood (see equation 4) of conditional video diffusion $p_\phi(\tau_x^i|w_i, x_{i,1})$ and inverse model $\prod_{t=1}^{T-1} p_\psi(a_{i,t}|x_{i,t}, x_{i,t+1})$. While sampling action trajectories and calculating their likelihoods during every step of the denoising process is computationally inefficient, we would still like to evaluate its effectiveness in visual plan refinements. However, we perform video diffusion in latent space while our inverse model is in observation space. Hence, for purpose of this experiment, we learn another inverse model $\overline{p_{\overline{\psi}}}(\tau_a^i|\tau_z^i)$ that uses a sequence model (i.e. a transformer) to produce an action trajectory $\tau_a^i$ given an observation trajectory in latent space $\tau_z^i$. We train $\overline{p_{\overline{\psi}}}$ for 20 epochs on 10k paired observation and action trajectories, each having a length of $T = 50$. We use AdamW optimizer, a batch size of 256 and a learning rate of $1e - 4$ during training. To generate an observation trajectory that maximizes the joint likelihood, we first sample 30 observation trajectories from video diffusion $p_\phi(\tau_x^i|w_i, x_{i,1})$ conditioned on subgoal $w_i$ and observation $x_{i,1}$. For each generated observation trajectory $\tau_x^i$, we sample a corresponding action trajectory $\tau_a^i$ and calculate its corresponding log-likelihood $\log \overline{p_{\overline{\psi}}}(\tau_a^i|v_{\text{enc}}(\tau_x^i))$. We select the observation trajectory $\tau_x^i$ with highest log-likelihood. Note that we only use $\overline{p_{\overline{\psi}}}$ for visual plan refinement and use $p_\psi$ for action execution to ensure fair comparison. If we use this approach for visual plan refinement with `HiP`, we obtain a success rate of $72.5 \pm 1.9$ on unseen tasks in `paint-block` environment. This is comparable to the performance of `HiP` with visual plan refinements from learned classifier $g_\psi$ which obtains a success rate of $72.8 \pm 1.7$ on unseen tasks in `paint-block` environment. In contrast, `HiP` without any visual plan refinement obtains a success rate of $71.1 \pm 1.3$ on unseen tasks in `paint-block` environment. These results show that $g_\psi$ serves as a good approximation for estimating whether an observation trajectory leads to a high-likelihood action trajectory, while still being computationally efficient.

**Architecture for $\overline{p_{\overline{\psi}}}$** We use a transformer model to represent $\overline{p_{\overline{\psi}}}(\tau_a|\tau_z = [\tau_z^T, \tau_z^H, \tau_z^W])$. We first use a ResNet-9 encoder to convert $\tau_z^T$, $\tau_z^H$, and $\tau_z^W$ to latent vectors. We then concatenate those latent vectors and project the resulting vector to a hidden space of 64 dimension using a linear layer. We then pass the 64 dimensional vector to a trajectory transformer model [23] which generates an action trajectory $\tau_a$ of length 50. The trajectory transformer uses a transformer architecture with 4 layers and 4 self-attention heads.

### E.3 How granularity of subgoals affects performance of `HiP` ?

We conduct a study in `paint-block` environment to analyze how granuality of subgoals affect `HiP`. In our current setup, a subgoal in `paint-block` domain is of form "Place <block color> block in/on/to <final block location>" and involves a pick and a place operation. We refer to our current setup as `HiP` (standard). We introduce two additional level of subgoal granuality:

- *Only one pick or place operation*: The subgoal will be of form "Pick <block color> block in/on <initial block location>" or "Place <block color> block in/on/to <final block location>". It will involve either one pick or one place operation. We refer to the model trained in this setup as `HiP` (more granular).

- *Two pick and place operations*: The subgoal will be of form "Place <1st block color> block in/on/to <final 1st block location> and Place <2nd block color> block in/on/to <final 2nd block location>". It will involve two pick and place operations. We refer to the model trained in this setup as `HiP` (less granular).

| Model | HiP (more granular) | HiP (Standard) | HiP (less granular) | UniPi |
|---|---|---|---|---|
| Paint-block (Seen) | $\mathbf{74.5 \pm 1.8}$ | $\mathbf{74.3 \pm 1.9}$ | $61.8 \pm 3.1$ | $37.2 \pm 3.8$ |
| Paint-block (Unseen) | $\mathbf{73.1 \pm 2.1}$ | $\mathbf{72.8 \pm 1.7}$ | $58.2 \pm 3.4$ | $35.3 \pm 3.2$ |

Table 2: **Granularity of Subgoals.** Performance of `HiP` as we vary the granularity of subgoals. Initially, it doesn't get affected but then starts to deteriorate when subgoals become too coarse.

Note that UniPi has the least granuality in terms of subgoals as it tries to imagine the entire trajectory from goal description. Table 2 in the rebuttal document compares `HiP` (standard), `HiP` (more granular), `HiP` (less granular) and UniPi on seen and unseen tasks in `paint-block` environment. We observe that `HiP` (standard) and `HiP` (more granular) have similar success rates where `HiP` (less granular) has a lower success rate. UniPi has the lowest success rate amongst these variants. We hypothesize that success rate of HiP remains intact when we decrease the subgoal granuality as long as the performance of visual planner doesn't degrade. Hence, `HiP` (standard) and `HiP` (more granular) have similar success rates. However, when the performance of visual planner degrades as we further decrease the subgoal granuality, we see a decline in success rate as well. That's why `HiP` (less granular) sees a decline in success rate and UniPi has the lowest success rate amongst all variants.

### E.4 Ablation on Visual Planning Model

| Model | Paint-block | | Object-arrange | | Kitchen-tasks | |
|---|---|---|---|---|---|---|
| | **Seen** | **Unseen** | **Seen** | **Unseen** | **Seen** | **Unseen** |
| HiP (RSSM) | $70.2 \pm 2.4$ | $69.5 \pm 1.6$ | $59.6 \pm 3.8$ | $59.2 \pm 3.9$ | $50.6 \pm 16.2$ | $46.8 \pm 19.4$ |
| HiP | $\mathbf{74.3 \pm 1.9}$ | $\mathbf{72.8 \pm 1.7}$ | $\mathbf{75 \pm 2.8}$ | $\mathbf{75.4 \pm 2.6}$ | $\mathbf{85.8 \pm 9.4}$ | $\mathbf{83.5 \pm 10.2}$ |

Table 3: **Ablating Visual Planner.** While performance gap between `HiP` and `HiP` (RSSM) is small in `Paint-block`, it widens in more visually complex domains, such as `Object-arrange` and `Kitchen-tasks`, thereby showing the importance of video diffusion model.

To show the benefits of video diffusion model, we perform an ablation where we use (text-conditioned) recurrent state space model (RSSM), taken from DreamerV3 [**?** ], as visual model for HiP. We borrow the RSSM code from [dreamerv3-torch codebase](#). To adapt RSSM to our setting, we condition RSSM on subgoal (i.e. subgoal encoded into a latent representation by Flan-T5-Base) instead of actions. Hence, sequence model of RSSM becomes $h_t = f(h_{t-1}, z_{t-1}, w)$ where w is latent representation of subgoal. Furthermore, we don't predict any reward since we aren't in a reinforcement learning setting and don't predict continue vector since we decode for a fixed number of steps. Hence, we remove reward prediction and continue prediction from the prediction loss. To make the comparisons fair, we pretrain RSSM with Ego4D data as well. We report the results in Table 3. We see that `HiP` with video diffusion model outperforms `HiP` with RSSM in all the three domains. While the performance gap between `HiP`(RSSM) and `HiP` (i.e. using video diffusion) is small in `paint-block` environment, it widens in `object-arrange` and `kitchen-tasks` domains as the domains become more visually complex.

### E.5 Analyzing sensitivity of iterative refinement to hyperparameters

| HiP | $\omega' = 0.5$ | $\omega' = 0.75$ | $\omega' = 1.0$ | $\omega' = 1.25$ | $\omega' = 1.5$ | $\omega' = 1.75$ | $\omega' = 2.0$ |
|---|---|---|---|---|---|---|---|
| Paint-block (Seen) | $71.8 \pm 2.3$ | $72.3 \pm 2.0$ | $\mathbf{74.3 \pm 1.9}$ | $\mathbf{73.9 \pm 2.2}$ | $72.1 \pm 1.7$ | $70.4 \pm 2.4$ | $68.2 \pm 1.9$ |
| Paint-block (Unseen) | $71.1 \pm 2.5$ | $71.4 \pm 1.8$ | $\mathbf{72.8 \pm 1.7}$ | $\mathbf{73.1 \pm 1.5}$ | $71.4 \pm 1.5$ | $69.3 \pm 2.7$ | $66.8 \pm 1.4$ |

Table 4: **Sensitivity of visual iterative refinement to guidance scale.** Performance of `HiP` as we vary the guidance scale $\omega'$. `HiP` performs best when $\omega' \in \{1, 1.25\}$ but performance degrades for higher values of $\omega'$.

The subgoal classifier doesn't introduce any test time hyperparameters and we use standard hyperparameters ($1e - 3$ learning rate, $1e - 6$ weight decay, 256 batch size, 50 epochs, Adam optimizer)

for its training which remains fixed across all domains. We observed that the performance changes minimally across different hyperparameters, given a learning rate decay over training. However, the observation trajectory classifier $g_\psi$ introduces an additional test time hyperparameter $\omega'$ which appropriately weights the gradient from observation trajectory classifier. Table 4 in the rebuttal document varies $\omega'$ between 0.5 and 2 in intervals of 0.25 and shows success rate of HiP. We see that HiP gives the best performance when $\omega' \in \{1, 1.25\}$ but it's performance degrades for higher values of $\omega'$.

## F  Analyzing Runtime of `HiP`

| Domain | Subgoal candidate generation | Subgoal classification | Visual planning per subgoal | Action planning per subgoal | Action execution per subgoal | Episodic runtime |
|---|---|---|---|---|---|---|
| Paint-block | 1.85s | 0.41s | 7.32s | 0.91s | 6.35s | 80.61 |
| Object-arrange | 1.9s | 0.43s | 7.39s | 0.89s | 9.57s | 78.71 |
| Kitchen-tasks | 1.81s | 0.41s | 7.35s | 0.98s | 1.28s | 40.37 |

Table 5: **Run time of** `HiP`. Average episodic run-time of `HiP`, along with average run-time of its different components for `Paint-block`, `Object-arrange` and `Kitchen-tasks` domains. While `HiP` has similar planning times across different domains, it has different action execution times and episodic runtimes across domains due to differences in simulation properties and average number of subgoals.

We provide average runtime of HiP for a single episode in all the three domains in Table 5 of the rebuttal document. We average across 1000 seen tasks in each domain. We break the average runtime by different components: task planning (subgoal candidate generation and subgoal classification), visual planning, action planning and action execution. We execute the action plan for a subgoal in open-loop and then get observation from the environment for deciding the next subgoal. From Table 5, we see that majority of the planning time is taken by visual planning. Recent works [24, 42, 56] have proposed techniques to reduce sampling time in diffusion models, which can be incorporated into our framework for improving visual planning speed in the future.

