# OpenReview forum: "Compositional Foundation Models for Hierarchical Planning"
_NeurIPS.cc/2023/Conference — NeurIPS 2023 poster_

### Official Review · Reviewer_zvNJ · 2023-07-07

**Soundness:** 3 good
**Presentation:** 3 good
**Contribution:** 2 fair
**Rating:** 5
**Confidence:** 4

**Summary:**

The paper presents Hierarchical Planning with Foundation Models (HiP), a framework that addresses long-horizon decision-making in novel environments, using multiple modalities of data and reasoning at various levels of hierarchy. The key elements of this framework involve the use of a large language model for constructing symbolic plans, a video diffusion model to generate observation trajectory plans, and a large pre-trained robot ego-centric action model to map these plans to a robot's control space. The authors propose an iterative refinement approach for feedback incorporation, promoting a consensus among different models, without the need for large model finetuning. Experimental results on two long-horizon tabletop manipulation tasks were presented, demonstrating promising results for the proposed strategy. The authors also mention the potential for including other modalities, like touch and sound, in the future.

**Strengths:**

The technical content of the paper appears to be sound, and the proposed framework has shown promising results in the experimental results provided. The methodology to combine language models, video diffusion models, and ego-centric robot control models into one system is comprehensive.
The paper is well-structured and clearly explains the approach and the reasoning behind the decisions made. The paper's flow from problem statement to proposed solution, and finally to experimental evaluation, is logical and easy to follow.

**Weaknesses:**

1. The baselines compared are all relatively weak. It's more appropriate to compare with foundation robotics models, such as SayCan, Gato[1], palm-e[2]. The authors argue that compared to saycan, they can generalise to new skills. However, they didn't demonstrate the generalization to new skills either. (They evaluated on unseen tasks but not new skills.)
2. While the iterative refinement approach is interesting, it could be further scrutinized to understand its limitations better, particularly concerning computational efficiency and robustness.
3. As this is mostly an experimental, please provide code to ensure reproducibility.
4. Formatting issue in line 9, 45.

[1] Scott Reed, Konrad Zolna, Emilio Parisotto, Sergio Gomez Colmenarejo, Alexander Novikov, Gabriel Barth-Maron, Mai Gimenez,
Yury Sulsky, Jackie Kay, Jost Tobias Springenberg, Tom Eccles, Jake Bruce, Ali Razavi, Ashley Edwards, Nicolas Heess, Yutian
Chen, Raia Hadsell, Oriol Vinyals, Mahyar Bordbar, and Nando de Freitas. A generalist agent. In Transactions on Machine Learning
Research (TMLR), November 10, 2022.
[2] Danny Driess, Fei Xia, Mehdi S. M. Sajjadi, Corey Lynch, Aakanksha Chowdhery, Brian Ichter, Ayzaan Wahid, Jonathan Tompson,
Quan Vuong, Tianhe Yu, Wenlong Huang, Yevgen Chebotar, Pierre Sermanet, Daniel Duckworth, Sergey Levine, Vincent Vanhoucke,
Karol Hausman, Marc Toussaint, Klaus Greff, Andy Zeng, Igor Mordatch, and Pete Florence. Palm-e: An embodied multimodal
language model. In arXiv preprint arXiv:2303.03378, 2023.

**Questions:**

See above. Could the authors discuss more about the efficiency and robustness? And also compare with a few foundation robotics model to better situate the work?

This paper presents a potentially significant step towards more effective long-horizon decision-making. The presented results are promising. I'm happy to raise my scores if the authors address my concerns.

**Limitations:**

The authors discussed the limited. They didn't comment on the broader societal impacts but I didn't see any concern.

---

> ### Author Rebuttal · Authors · 2023-08-10
>
> We thank reviewer zvNJ for their constructive feedback. We now answer the following concerns raised in the review.
>
> >The baselines compared are all relatively weak. It's more appropriate to compare with foundation robotics models, such as SayCan, Gato[1], palm-e[2]. The authors argue that compared to saycan, they can generalise to new skills. However, they didn't demonstrate the generalization to new skills either. (They evaluated on unseen tasks but not new skills.)
>
> We add comparisons to 2 existing foundation models for decision making: Gato [1] and Say-Can [2]. In training the visual and action planning for HiP, we use 100k robot videos for visual planner and the inverse dynamics, when trained from scratch, utilizes 10k state-action trajectory pairs. In order to ensure fair comparison, we use 110k datapoints for training Gato and Say-Can.
>
> *Gato:* We borrow the Gato architecture from [Vima Codebase](https://github.com/vimalabs/VIMA/tree/main) and use it for training a language conditioned policy with imitation learning. We use 110k (langauge, observation trajectory, action trajectory) datapoints in each of the three domains for training Gato. Furthermore, we provide oracle subgoals to Gato. We compare Gato with HiP and other baselines on all three domains in Table 1 of the rebuttal document. We see that HiP outperforms Gato on both seen and unseen tasks in all of the three domains.
>
> *SayCan:* We borrow the SayCan algorithm from the [SayCan codebase](https://github.com/google-research/google-research/blob/master/saycan/SayCan-Robot-Pick-Place.ipynb) and adapt it to our settings. Following the recommendations of SayCan codebase, we use CLIPort policies as primitives. CLIPort policies take in top-down RGBD view and outputs pick and place pixel coordinates. Then, an underlying motion planner picks the object from the specified pick-coordinate and places the object at the specified place-coordinate. We train CLIPort policies on 110k (language, observation, action) datapoints in *paint-block* and *object-arrange* domain. The SayCan paper uses value function as an affordance function to select the correct subgoal given current observation and high level goal. However, CLIPort policies don't have a value function. The SayCan codebase uses a hardcoded scoring function which doesn't apply to *object-arrange* domain. To overcome these issues, we use the LLM grounding strategy from Huang et al. [3]. It uses unnormalized logits over the pixel space given by CLIPort policies as affordance and uses it to ground LLM to current observation and thus predict the subgoal. We then compare SayCan with HiP and other baselines on *paint-block* and *object-arrange* domain in Table 1 of the rebuttal document. While SayCan outpeforms other baselines, HiP still outperforms it both on seen and unseen tasks of *paint-block* and *object-arrange* domain. We couldn't run SayCan on *kitchen-tasks* domain as there's no clear-cut primitive in that domain. This points to a limitation of SayCan which requires tasks to be expressed in terms of primitives with each primitive paired with an affordance function.
>
> PaLM-E[4] is based on finetuning PaLM models and associated vision encoders on robotics data. However, PaLM models aren't available and finetuning a large LLM (540B parameters) along with a vision encoder (22B parameters) requires a large number of GPUs that aren't available in an academic lab.
>
> >As this is mostly an experimental, please provide code to ensure reproducibility.
>
> We have uploaded anonymized preliminary version of code. We also provided architectural details in Appendix B of our submission and further details on training and evaluation procedure in Appendix C of our submission. We will work on cleaning up the code over coming weeks and provide a clean version of the code with detailed instructions with the camera ready version of the paper.
>
> >While the iterative refinement approach is interesting, it could be further scrutinized to understand its limitations better, particularly concerning computational efficiency and robustness.
>
> **Computational efficiency of iterative refinement:** We provide average runtime of HiP for a single episode in all the three domains in Table 2 of the rebuttal document. As we can see, visual planning takes majority of the runtime. The prediction time by subgoal classifier $f_\phi$ is minimal and gradients from observation trajectory classifier $g_\psi$ adds minimal runtime overhead. However, the subgoal classifier and observation trajectory classifier adds memory overhead.
>
> **Robustness of iterative refinement:** By robustness, we assume you are referring to sensitivity analysis with respect to hyperparameters. The subgoal classifier doesn't introduce any test time hyperparameters and we use standard hyperparameters (1e-3 learning rate, 1e-6 weight decay, 256 batch size, 50 epochs, Adam optimizer) for its training which remains fixed across all domains. We observed that the performance changes minimally across different hyperparameters, given a learning rate decay over training. However, the observation trajectory classifier $g_\psi$ introduces an additional test time hyperparameter $\omega'$ which appropriately weights the gradient from observation trajectory classifier. Table 4 in the rebuttal document varies $\omega'$ between 0.5 and 2 in intervals of 0.25 and shows success rate of HiP. We see that HiP gives the best performance when $\omega' \in \{1, 1.25\}$ but it's performance degrades for higher values of $\omega'$. We will include this analysis in camera ready version of our paper.
>
>
> **References:**\
> [1] Reed et al. "A Generalist Agent." TMLR, 2022.\
> [2] Ahn et al. "Do As I Can, Not As I Say: Grounding Language in Robotic Affordances." CoRL, 2022.\
> [3] Huang et al. "Grounded Decoding: Guiding Text Generation with Grounded Models for Robot Control." arXiv:2303.00855, 2023.\
> [4] Driess et al. "PaLM-E: An Embodied Multimodal Language Model." ICML, 2023.

---

> > ### Author Response · Authors · 2023-08-16
> >
> > Dear reviewer zvNJ,
> >
> > Thank you again for your comments and suggestions on our paper. We hope that our responses and new results have addressed your questions and concerns. We still have a few days left in the discussion period. If you have any further questions, please don't hesitate to let us know and we'll be happy to address them. Thank you!
> >
> > Best,
> >
> > Authors

---

> > > ### Comment · Reviewer_zvNJ · 2023-08-20
> > >
> > > I would like to thank the authors for their detailed response. Most of my concerns have been addressed, and I have updated my score.

---

### Official Review · Reviewer_z2ah · 2023-07-07

**Soundness:** 3 good
**Presentation:** 3 good
**Contribution:** 3 good
**Rating:** 7
**Confidence:** 3

**Summary:**

This work presents a new approach of combining foundation models from different modalities to into hierarchical system to solve long-horizon robotics problem. The performance of the system is evaluated on two long-horizon tasks and largely out-performed other baselines.

**Strengths:**

The approach of this work is a effective combination of previously available models. The effectiveness of iterative refinement is particularly interesting (which seems to be one of the main performance booster in this approach). The framework details are well supported by the writing and supplementary materials.

**Weaknesses:**

I don't find weakness of this work, but I do have some questions in terms of the dataset, experiment, and analysis, see questions below.

**Questions:**

1. What does dataset stats look like?
2. Can you provide more description rationalize the use of training data to evaluate HiP (Table 1)?
3. line 213: What is benefits of pre-training video diffusion model in terms of success rate? The description seems to be only indirectly describing the benefits?
4. It would be interesting to see some analysis on the effectiveness of subgoal generation in this approach, for example, if the granularity of the subgoal generation various, what effect does it have on the overall approach. (This is interesting to me, without this will not affect my review score.)
5. How is the computation speed look like for this approach, how fast it is to generate for an episode?

**Limitations:**

Limitations are addressed in the paper.

---

> ### Author Rebuttal · Authors · 2023-08-10
>
> We thank reviewer z2ah for evaluating our work. We now answer the following concerns raised in the review.
>
> >What does dataset stats look like?
>
> We described our evaluation environments in Section 3.1 of our original submission and provided further details about our datasets for task planning, visual planning and action planning in Appendix C of our original submission. We now summarize these dataset statistics here:
> 1. Task planning:
> 	 (a) LLM prompting: For all the three domains, we prompt the LLM with 5 examples of (high level goals, set of desirable subgoals) pairs sampled from tasks in given domain.
> 	 (b) Training multi-class classifier $f_\phi$ for subgoal classification: We used classification dataset $D_{classify}$ , consisting of observation $x_{i,1}$, goal g, candidate subgoals $(w_j)\_{j=1}^M$ and the correct subgoal label i, for training multi-class classifier $f_\phi$. The classification dataset for *paint-block*, *object-arrange*, and *kitchen-tasks* consists of 58k, 82k and 50k datapoints respectively.
>
> 2. Visual planning: After pretraining our visual model on Ego4D dataset consisting of 344k datapoints, we finetune our visual model on dataset $D_{video} \coloneqq (\tau_x^i, w_i)$ consisting of (observation trajectory, subgoal) pairs. The video dataset consisted of 100k datapoints for all the domains.
>
> 3. Action planning: We train VC-1 [1] initialized inverse dynamics model on dataset $D_{\text{inv}} \coloneqq (\tau_x^i, \tau_s^i)$ consisting of paired observation and robot state trajectories. The inv dataset consisted of 1k datapoints on *paint-block* and *object-arrange* domains and 3.5k datapoints on *kitchen-tasks* domain.
>
> >Can you provide more description rationalize the use of training data to evaluate HiP (Table 1)?
>
> We are not sure if we understand this question completely, but how we understood it is that you are asking about the evaluation metrics used in Table 1. We measure the task completion rate (i.e. whether the final goal has been achieved) for evaluating HiP and other baselines on *paint-block* and *object-arrange* domains. To be consistent with evaluation metrics used in [1,2], we use subtask completion rate (i.e. percentage of subtasks completed) for evaluating HiP and other baselines on *kitchen-tasks* domain. To evaluate a model on Seen and Unseen tasks, we sample 1000 tasks from $T_{train}$ and $T_{test}$ respectively, and obtain average task completion rate on *paint-block* and *object-arrange* domains and average subtask completion rate on *kitchen-tasks* domain. We repeat this procedure over 4 different seeds and report the mean and the standard error over those seeds in Table 1. We will update our draft to better describe our evaluation metrics in camera ready version of the paper. We apologize if we misinterpreted your question and we would be happy to offer further clarifications.
>
> >What is benefits of pre-training video diffusion model in terms of success rate? The description seems to be only indirectly describing the benefits?
>
> Figure 2 in the rebuttal document provides success rate of HiP when video diffusion model is trained with 100% training data, 75% training data and 50% training data, with or without Ego4D [2] pretraining. We see that pretraining with Ego4D consistently yields better success rate even with reduced training dataset sizes.
>
> >It would be interesting to see some analysis on the effectiveness of subgoal generation in this approach, for example, if the granularity of the subgoal generation various, what effect does it have on the overall approach.
>
> We appreciate that the reviewer suggested this interesting axis to explore. So, we conduct a study in *paint-block* domain to analyze how granuality of subgoals affect HiP. In our current setup, a subgoal in *paint-block* domain is of form "Place <block color> block in/on/to <final block location>" and involves a pick and a place operation. We refer to our current setup as HiP (standard). We introduce two additional level of subgoal granuality:
> 1. Only one pick or place operation: The subgoal will be of form "Pick <block color> block in/on <initial block location>" or "Place <block color> block in/on/to <final block location>". It will involve either one pick or one place operation. We refer to the model trained in this setup as HiP (more granular).
> 2. Two pick and place operations: The subgoal will be of form "Place <1st block color> block in/on/to <final 1st block location> and Place <2nd block color> block in/on/to <final 2nd block location>". It will involve two pick and place operations. We refer to the model trained in this setup as HiP (less granular).
>
> Note that UniPi has the least granuality in terms of subgoals as it tries to imagine the entire trajectory from goal description. Table 3 in the rebuttal document compares HiP (standard), HiP (more granular), HiP (less granular) and UniPi on seen and unseen tasks in *paint-block* domain. We observe that HiP (standard) and HiP (more granular) have similar success rates where HiP (less granular) has a lower success rate. UniPi has the lowest success rate amongst these variants. We hypothesize that success rate of HiP remains intact when we decrease the subgoal granuality as long as the performance of visual planner doesn't degrade. Hence, HiP (standard) and HiP (more granular) have similar success rates. However, when the performance of visual planner degrades as we further decrease the subgoal granuality, we see a decline in success rate as well. That's why HiP (less granular) sees a decline in success rate and UniPi has the lowest success rate amongst all variants.
>
> **References:**\
> [1] Majumdar et al. "Where are we in the search for an Artificial Visual Cortex for Embodied Intelligence?" arXiv:2303.18240, 2023.\
> [2] Grauman et al. "Ego4D: Around the World in 3,000 Hours of Egocentric Video." CVPR, 2022.

---

> > ### Author Response · Authors · 2023-08-16
> >
> > Dear reviewer z2ah,
> >
> > Thank you again for your comments and suggestions on our paper. We hope that our responses and new results have addressed your questions and concerns. We still have a few days left in the discussion period. If you have any further questions, please don't hesitate to let us know and we'll be happy to address them. Thank you!
> >
> > Best,
> >
> > Authors

---

> > ### Comment · Reviewer_z2ah · 2023-08-17
> >
> > Thanks for addressing my questions! I think this work makes good contribution towards combining models and robot planning. I will keep my score.

---

### Official Review · Reviewer_3JTC · 2023-07-07

**Soundness:** 3 good
**Presentation:** 3 good
**Contribution:** 2 fair
**Rating:** 4
**Confidence:** 3

**Summary:**

The paper introduces a framework called Hierarchical Planning with Foundation Models (HiP), which is designed to improve decision-making in new environments with long-term goals. HiP uses hierarchical reasoning to plan subgoals, visually reason about plans, and execute actions through visual-motor control. The framework utilizes different kinds of knowledge to support various levels of decision-making. It employs a large language model for constructing symbolic plans, a large video diffusion model for grounding the plans in the environment, and an inverse dynamics model for inferring actions from generated videos. The models are kept consistent through iterative refinement. The effectiveness of HiP is demonstrated through its application in two long-horizon table-top manipulation tasks.

The paper's introduction explains that successful execution of tasks in unfamiliar environments requires reasoning at abstract, geometric, and control levels. The authors propose using multiple modalities of Internet-scale data to reason across these levels. They explain the limitations of existing models and propose HiP as a solution, which uses three different large foundation models to construct a physically executable plan for long-horizon tasks. The models are iteratively refined based on feedback from downstream models, ensuring that the final plan satisfies constraints at all levels. The authors also discuss the potential of training foundation models for videos and ego-centric actions, and the limitations of their current computational resources. They conclude by highlighting the promise of their proposed approach in long-horizon decision-making tasks and hope to inspire future work in more complex real-world tasks.

**Strengths:**

- Hierarchical Planning with Foundation Models (HiP) leverages different modalities of knowledge, which can improve decision-making in novel environments with long-horizon goals.
- The framework uses a large language model, a large video diffusion model, and an inverse dynamics model, allowing it to construct symbolic plans, ground those plans in the environment, and infer actions from generated videos.
- The iterative refinement process ensures consistency between the models and enables hierarchically consistent plans that are responsive to the goal and executable given the current state and agent.
- The approach is computationally efficient to train as it doesn't require any large model finetuning.
- The authors demonstrate promising results on two long-horizon tabletop manipulation environments, illustrating the efficacy and adaptability of the approach.

**Weaknesses:**

- The paper is technically similar to UniPi. The technical novelty could be potentially incremental - it adds language model to decompose the language goal to language subgoals and then apply video prediction model to it.
- The work relies on foundation models for video prediction and ego-centric action prediction. Concern: Although it can be expected that video prediction model will be available in the future, but it may not be assumed to be suitable for robotics applications. Additionally, it seems to be overclaim that robotics should rely on these models.
    1. The applications in robotics may need to model multiple aspects of physical environments, where “expected” video foundation models can struggle to learn
    How could forces be modeled in video model? How to guarantee the generated video are feasible in physics and can be executed for unseen tasks? The shown pick-and-place is a useful environment, but doesn’t support all the claims. Either the claims need to be revised or more experiments are needed.
    2. It is also likely that vision (image/video) is only one of several important modalities in real-world robotic decision-making. For example, for some insertion tasks, tactile sensing could be important.
    3. The computational cost of these models is not known yet at all. Although GPT4 is available for use, it only needs to transfer text over internet. If the needed models were to transfer high-quality images or even videos, how much bandwidth would they need? Is that possible to allow real-time robotic decision-making? It seems unlikely that in the near future we will have pretrained action/video models for all scenarios that can give real-time results over internet. If the goal of this paper is to explore along this direction, I think these aspects should all be considered, instead of assuming all these challenges are nonexistent and just claim the contributions.
    - Overall, I think the paper is a meaningful initial exploration towards this direction, but the claims of the paper should be clear about this as some assumptions are not necessarily realistic. The title “Hierarchical Planning with Foundation Models” does not necessarily match the contributions and novelties of this paper.
- L68 — The step 2 “visual planning” is the key step to plan in physical level, but image-level is not necessarily the correct level of abstraction for many cases. When does it apply and when not? This seems to include lots of unnecessary details of the physical world for tasks other than just tabletop manipulation.
- More general concern: The approach relies on the availability of large pretrained models in different domains. Currently, these models are only readily available in the language domain. Furthermore, as mentioned in the paper, the approach is demonstrated on smaller-scale video and ego-centric action models trained in simulation, which serve as proxies for larger pretrained models. This might limit the generalizability of the results to more complex real-world tasks.

**Questions:**

See the weaknesses.

**Limitations:**

Yes

---

> ### Author Rebuttal · Authors · 2023-08-10
>
> We thank reviewer 3JTC for their constructive feedback. We now answer the following concerns raised in the review.
>
> >Novelty
>
> Prior works using foundation models for robotics typically learn a single large model on Internet/robotics data. However, LLMs cannot construct visually grounded plans and UniPi cannot construct long horizon plans. The novelty of approach is to use a hierarchy of different foundation models in combination, to hierarchically reason across different modalities, enabling us to scale to long-horizon plans. This hierarchy further allows us to use passive pre-training on different modalities of data, significantly reducing the need for domain-specific data. We further propose to use iterative refinement to connect models across different hierarchies together and ensure compatability across modalities. We demonstrate our results on three long-horizon robot manipulation domains.
>
> >Use Case of Video Models in Robotics
>
> Video prediction models trained on Internet data can provide us with a rich source of motion information and physics. Given a text subgoal and current image observation, a synthesized video can tell us the precise hand motions necessary to open a door. Our submission shows that video models pretrained on Internet data (Ego4D) generate better videos (in terms of FVD; see Figure 5 in our submission) and lead to higher success rate (see Figure 2 in the rebuttal document). As a result, while large video prediction models do not eliminate the need for robotics data, we believe they provide a rich prior of semantic and physics information that reduce the amount of in-domain robotics data needed, by teaching us priors on how objects move and their physics. We will clarify this claim and accordingly update our introduction.
>
> > How to model forces and tactile sensing?
>
> We agree that the current instantiation does not integrate force and tactile sensing and will discuss it in our limitations. However, we note that a video model can be used for both force and tactile sense. For instance, [2] proposed a visual model to predict future visual observations and gelsight observations given current visual and gelight observations and planned actions. Building on this work, we can extend our visual planner to additionally generate future gelsight observations given a subgoal. Then, these predicted gelsight observations can be used as input to our inverse dynamics model. While our submission doesn't focus on touch sensing, we believe future works that build foundation models for it can easily be integrated. We will update our Section 5 to discuss the limitations of our current instantiation of hierarchy and how our framework can be expanded to include additional sensory modality like touch.
>
> > Computational Cost / Bandwith of Video
>
> **Bandwidth Cost of Images/Videos:** In our submission, we generate videos of size 64x48 consisting of 50 frames, which amount to ~15Kb. Even if we generate 200 frames at a resolution of 256x256, they would still be under 1MB. Hence, we do not expect the transfer of these data to become a bottleneck. Also, by generating a video and action plan for each subgoal instead of at each timestep, we further reduce the bandwidth requirement. However, this approach may not be suitable for dynamically changing environments that generate subgoals at a high frequency and will discuss this limitation in Section 5 in camera ready version.
>
> **Computational Cost of Running Models:** We provide average runtime of HiP for a single episode in all the three domains in Table 2 of the rebuttal document. Please see our common response for more details. We ran all our inference experiments on A6000 which is more powerful than on-board GPUs available on robotics platform. However, we believe that improvements in both hardware and software (e.g. quantization techniques for faster inference) will help realize our models on on-board GPUs.
>
> >  Claim of paper / Contributions
>
> As discussed in the comments above, we will clarify some assumptions that our approach makes (eg: video models serve as a prior and doesn't neccassarily eliminate the need for in-domain robotics data), and further expand upon the limitations of our work in Section 5 (e.g.: the scalability when needed to generate new subgoals at a high frequency) and tone down our overall claims.
>
> Our work focuses on leveraging hierarchies for long-horizon decision making with each level of hierarchy making use of a foundational model pre-trained on non-robotics data. Hence, we titled our paper as "Hierarchical Planning with Foundation Models". However, we are happy to modify as the reviewer sees fit.
>
> >L68 — Images as Abstraction of Physical World
>
> In general, we agree that image-level may not be the most efficient  abstraction for the physical planning and can capture unnecessary detail. In settings in which there is no ego-motion, which corresponds to many manipulation tasks, then image-level motion corresponds well to physical movement. In other settings with ego-motion, then image-level plans may be unnecessary details from ego-motion, but will still also include the relevant physical motion. However, we note that image-level representation is still sufficient for this setting, and the best text-conditioned video generation models can capture relatively accurate physical motion even with ego-motion[3]. We did a study where we replaced our video diffusion model with a latent RSSM, and modeled physical-level plans in the latent space. As seen in Table 1 of the rebuttal document, video generation outperforms this baseline, with larger performance gap with more visually complex domains
>
> **References**:
> [1] Majumdar et al. "Where are we in the search for an Artificial Visual Cortex for Embodied Intelligence?" arXiv, 2023.
> [2] Calandra et al. "More than a feeling: Learning to grasp and regrasp using vision and touch.", RAL, 2018.
> [3] Ho et al. "Imagen Video: High Definition Video Generation with Diffusion Models" arXiv,2022.

---

> > ### Author Response · Authors · 2023-08-16
> >
> > Dear reviewer 3JTC,
> >
> > Thank you again for your comments and suggestions on our paper. We hope that our responses and new results have addressed your questions and concerns. We still have a few days left in the discussion period. If you have any further questions, please don't hesitate to let us know and we'll be happy to address them. Thank you!
> >
> > Best,
> >
> > Authors

---

### Official Review · Reviewer_7a1H · 2023-07-11

**Soundness:** 3 good
**Presentation:** 3 good
**Contribution:** 3 good
**Rating:** 6
**Confidence:** 4

**Summary:**

This paper presents a way to leverage large foundation models to perform long-horizon hierarchical planning tasks. Specifically, the authors leverage an LLM for subgoal generation from the text instruction, a video generation model for generating a plan for each subgoal, and an action prediction model that outputs an action given the current and next observation generated by the video generation model. To make the predictions between different models consistent, the authors propose the iterative refinement procedure to refine the prediction of a model based on joint distributions. The paper includes results in simulated manipulation environments and shows improvement over prior goal-conditioned/action planner/video planner-based methods. The authors also include ablation studies to demonstrate effectiveness of various design choices in the method, such as the use of iterative refinement.

**Strengths:**

1. The authors presents a nice way to combine three different large foundation models in the problem of hierarchical planning. The iterative refinement method seems very effective in aligning predictions of various models, which is of great significance in research that plays with using multiple foundation models.

2. The authors perform rigorous empirical evaluations via comparing to various baselines and conducting a large number of ablation studies to show the good performance of the proposed method.

**Weaknesses:**

1. While iterative refinement method is intuitively reasonable and a natural choice for making predictions of models in the pipeline consistent, the authors use approximations of the target joint distribution in the iterative refinement step. It is unclear if the some approximations fully capture the target distribution. For example, I'm not sure if Eq. (3) is a good approximation of (2) since Eq. (3) doesn't have the reachability part and thus doesn't fully capture the feasibility of achieving the language goal.

2. It is unclear if the method can outperform previous hierarhical planning methods such as [1, 2, 3, 4, 5, 6]. Since the authors only evaluate the method in simple simulated pick-and-place tasks, I'm not sure if we need large-scale video diffusion models for video generation and rather simple latent-space world model like [1] is very likely to work.

3. Following the comment in 2, I think the authors should evaluate the method in more domains included in [1,2,3,4,5,6] and also more realistic domains such as real-world robot manipulation settings to fully demonstrate the necessity of using big video foundation models.

[1] Hafner, Danijar, Kuang-Huei Lee, Ian Fischer, and Pieter Abbeel. "Deep hierarchical planning from pixels." Advances in Neural Information Processing Systems 35 (2022): 26091-26104.

[2] Hafner, Danijar, Jurgis Pasukonis, Jimmy Ba, and Timothy Lillicrap. "Mastering diverse domains through world models." arXiv preprint arXiv:2301.04104 (2023).

[3] Zhao, Chao, Shuai Yuan, Chunli Jiang, Junhao Cai, Hongyu Yu, Michael Yu Wang, and Qifeng Chen. "ERRA: An Embodied Representation and Reasoning Architecture for Long-horizon Language-conditioned Manipulation Tasks." IEEE Robotics and Automation Letters (2023).

[4] Kujanpää, Kalle, Joni Pajarinen, and Alexander Ilin. "Hierarchical Imitation Learning with Vector Quantized Models." arXiv preprint arXiv:2301.12962 (2023).

[5] Mendonca, Russell, Oleh Rybkin, Kostas Daniilidis, Danijar Hafner, and Deepak Pathak. "Discovering and achieving goals via world models." Advances in Neural Information Processing Systems 34 (2021): 24379-24391.

[6] Tian, Stephen, Suraj Nair, Frederik Ebert, Sudeep Dasari, Benjamin Eysenbach, Chelsea Finn, and Sergey Levine. "Model-based visual planning with self-supervised functional distances." arXiv preprint arXiv:2012.15373 (2020).

**Questions:**

Please address the concerns raised in the section above.

Update after rebuttal:

I decided to raise my score after reading author’s response.

**Limitations:**

Yes.

---

> ### Author Rebuttal · Authors · 2023-08-10
>
> We thank reviewer 7a1H for their constructive feedback. We now answer the following concerns raised in the review.
>
> >unclear if the some approximations fully capture the target distribution. For example, I'm not sure if Eq. (3) is a good approximation of (2) since Eq. (3) doesn't have the reachability part and thus doesn't fully capture the feasibility of achieving the language goal.
>
> In Appendix D.1 of our supplemental material, we compare with optimizing the correct target distribution compared to our approximations. We find that directly optimizing Equation (2) gets an average success rate of $54.3 \pm 7.2$ compared to optimizing  Equation (3) which gets a performance of $98.2 \pm 1.5$ on *paint-block* domain. This suggests that our approximations reasonably capture the target distribution.
>
> Furthermore, if we consider the rewrite of Equation (3),
>
> $\max_{w_i} \log p_\text{LLM}(w_i|g) + \log\left(\frac{p(x_{i,1} | w_i, g)}{p(x_{i,1} | g)}\right)$
>
> we see that the first part $\log p_\text{LLM}(w_i|g)$ ensures reachability as it makes LLM produce subgoals that make progress towards the high-level goal and the second part $\log\left(\frac{p(x_{i,1} | w_i, g)}{p(x_{i,1} | g)}\right)$ ensures feasibility as the classifier approximating the log density ratio selects subgoals feasible from $x_{i,1}$.
>
>
> >It is unclear if the method can outperform previous hierarhical planning methods such as [1, 2, 3, 4, 5, 6]. Since the authors only evaluate the method in simple simulated pick-and-place tasks, I'm not sure if we need large-scale video diffusion models for video generation and rather simple latent-space world model like [1] is very likely to work.
>
> To show the benefits of video diffusion model, we perform an ablation where we use (text-conditioned) recurrent state space model (RSSM), taken from Dreamer-v3 [2] (and used in other prior works [1,3]), as visual model for HiP. We borrow the RSSM code from [dreamerv3-torch codebase](https://vitalab.github.io/article/2023/01/19/DreamerV3.html). To adapt RSSM to our setting, we condition RSSM on subgoal (i.e. subgoal encoded into a latent representation by Flan-T5-Base) instead of actions. Hence, sequence model of RSSM becomes $h_t = f(h_{t-1}, z_{t-1}, w)$ where w is latent representation of subgoal. Furthermore, we don't predict any reward since we aren't in a reinforcement learning setting and don't predict continue vector since we decode for a fixed number of steps. Hence, we remove reward prediction and continue prediction from the prediction loss. To make the comparisons fair, we pretrain RSSM with Ego4D data as well. We report the results in Table 1 of the rebuttal document. We see that HiP the video diffusion model outperforms HiP with RSSM in all the three domains. While the performance gap between HiP(RSSM) and HiP (i.e. using video diffusion) is small in *paint-block* domain, it widens in *object-arrange* and *kitchen-tasks* domains as the domains become more visually complex. Note that we didn't directly compare to Dreamer-v3 as Dreamer-v3 operates in online RL setting. Since we wanted to show benefits of using video diffusion, we took RSSM from Dreamer-v3 and used it with HiP for a fair comparison. We will add this ablation study on choice of video model in camera ready version of our paper
>
> >Following the comment in 2, I think the authors should evaluate the method in more domains included in [1,2,3,4,5,6] and also more realistic domains such as real-world robot manipulation settings to fully demonstrate the necessity of using big video foundation models.
>
> We evaluate HiP on a third domain *kitchen-tasks*, inspired by the *kitchen-shift* domain in Xing et al. [4], and is a generalization of *Robokitchen* domain used in Mendoca at al. [3]. Please see our common response for more details on the task setup and results.
>
>
> **References**\
> [1] Hafner et al. "Deep hierarchical planning from pixels." NeurIPS, 2022.\
> [2] Hafner et al. "Mastering diverse domains through world models." arXiv, 2023.\
> [3] Mendonca et al. "Discovering and achieving goals via world models." NeurIPS, 2021.\
> [4] Xing et al. "KitchenShift: Evaluating Zero-Shot Generalization of Imitation-Based Policy Learning Under Domain Shifts."

---

> > ### Author Response · Authors · 2023-08-16
> >
> > Dear reviewer 7a1H,
> >
> > Thank you again for your comments and suggestions on our paper. We hope that our responses and new results have addressed your questions and concerns. We still have a few days left in the discussion period. If you have any further questions, please don't hesitate to let us know and we'll be happy to address them. Thank you!
> >
> > Best,
> >
> > Authors

---

> > ### Comment · Reviewer_7a1H · 2023-08-20
> > **Reply to rebuttal**
> >
> > Thank you for the rebuttal! I appreciate the additional experiments and clarifications, which addresses most of my concerns. One question is why directly optimizing the correct target distribution (Eq. (2)) leads to much poorer performance. Is there something off? Overall, I’m satisfied with author’s response and would like to raise my score.

---

> > > ### Author Response · Authors · 2023-08-20
> > >
> > > Thank you for your time reviewing the paper. Please see our clarification:
> > >
> > > >One question is why directly optimizing the correct target distribution (Eq. (2)) leads to much poorer performance. Is there something off?
> > >
> > > We believe that the main reason optimizing the target distribution (Eq. (2)) leads to poor performance is because each of the underlying learned probability densities do not fully accurately capture each probability distribution in (Eq. (2)). For instance, DDPMs are typically trained with loss weightings to preferentially generate high sample quality samples as opposed to accurately modeling the underlying probability distribution. As models get larger, each learned probability density will more accurately model each probability distribution in Eq. 2. and we believe that then the performance with significantly rise. We will update Appendix D.1 to better clarify this point in the camera-ready version of our paper.

---

### Author Rebuttal · Authors · 2023-08-10

We thank the reviewers for their thoughtful suggestions. We want to start by addressing the common concerns brought up by the reviewers and dive into the remaining points in the individual responses.

### **Evaluation on more realistic robotic domain**:

We evaluate HiP on an additional third domain kitchen-tasks, inspired by the kitchen-shift domain in Xing et al. [1], and is a generalization of Robokitchen domain used in Mendoca et al. [2]. Description of kitchen-tasks domain: A robot has to complete kitchen subtasks given in language goal instructions, such as open microwave and light the kitchen area. However, the environment may have objects irrelevant to the subtasks that the robot must ignore. Furthermore, some kitchen subtasks may already be completed, and the robot needs to ignore those tasks when completing the goal. There are 7 possible kitchen subtasks: opening the microwave, moving the kettle, switching on lights, turning on the bottom knob, turning on the top knob, opening the left drawer, and opening the right drawer. A new task T is generated by constructing a random sequence of 4 out of 7 possible kitchen subtasks. In generating the random sequence, we randomly select an instance of microwave out of 3 possible instances, an instance of kettle out of 4 possible instances, a texture of counter, floor, and drawer independently out of 3 possible textures and randomizing the initial poses of the kettle and microwave. With 50% probability, one of 4 selected kitchen subtask is completed before the start of the task. Hence, tasks usually have 3~4 subtasks (i.e. subgoals).

**Results**: We evaluate HiP and new baselines on kitchen-tasks in Table 1 of the rebuttal document. We use subtask completion rate as our evaluation metric for kitchen-tasks domain to be consistent with evaluation metric used in [1,2]. We see that HiP outperforms all other baselines significantly on both seen and unseen tasks in kitchen-tasks domain.

### **New Baselines**:

We add comparisons to 2 existing foundation models for decision making: Gato and Say-Can. We compare these against these new baselines on our experiment domains in Table 1 of the rebuttal document. We see that HiP outperforms Gato and Say-Can on both seen and unseen tasks.

Say-Can specifically has an inherent limitation in that it requires tasks to be expressed in terms of primitives, each of which is paired with an affordance function. However, CLIPort policies don't have a value function. To overcome this issue in our experiment domains, we use the LLM grounding strategy from Huang et al. [9].

### **Ablation on Video Model**:

To show the benefits of video diffusion model, we perform an ablation where we use (text-conditioned) recurrent state space model (RSSM) ,taken from Dreamer-v3 [10], as visual model for HiP. We report the results in Table 1 of the rebuttal document. We see that HiP the video diffusion model outperforms HiP with RSSM in all the three domains. It is evident that the performance gap increases in *object-arrange* and *kitchen-tasks* domains, where the domains are more visually complex.

### **Runtime characteristics of HiP**:

 We provide average runtime of HiP for a single episode in all the three domains in Table 2 of the rebuttal document. We average across 1000 seen tasks in each domain. We break the average runtime by different components: task planning (subgoal candidate generation and subgoal classification), visual planning, action planning and action execution. We execute the action plan for a subgoal in open-loop and then get observation from the environment for deciding the next subgoal. From Table 2, we see that majority of the planning time is taken by visual planning.

### **Technical contributions**:

Our submission makes following technical contributions:

(i) Proposing a hierarchical task planning framework for long-horizon decision-making, divided into high-level task planning in text, visual planning with videos, and action planning with joint state trajectories

(ii) Designing an iterative refinement strategy to ensure consistency amongst the different levels of planning

(iii) Demonstrating that pre-training each level of hierarchy on different modalities of easily accessible Internet data with small amounts of domain-specific data can be effective.

In summary, we propose a general robot planning strategy that leverages a hierarchy of foundation models, which can learned separately on different modalities of Internet and robotics data, to construct long-horizon plans. This is different from existing works leveraging foundation models, with typically use a single model (typically a LLM or video model) to make decisions, and allows our approach to reason hierarchically across different modalities, enabling us to scale better to long-horizon plans. We illustrate the efficacy of our results on three long-horizon robot manipulation domains.

**References**:

[1] Xing et al. "KitchenShift: Evaluating Zero-Shot Generalization of Imitation-Based Policy Learning Under Domain Shifts."

[2] Mendonca et al. "Discovering and achieving goals via world models." NeurIPS, 2021.

[3] Salimans et al. "Progressive Distillation for Fast Sampling of Diffusion Models.", ICLR 2022.

[4] Zheng et al. "Fast Sampling of Diffusion Models via Operator Learning", arXiv:2211.13449, 2023.

[5] Janner et al. "Planning with Diffusion for Flexible Behavior Synthesis." ICML 2022.

[6] Ho et al. "Imagen Video: High Definition Video Generation with Diffusion Models" arXiv,2022.

[7] Yu et al."Video Probabilistic Diffusion Models in Projected Latent Space." arXiv, 2023.

[8] Singer et al. "Make-A-Video: Text-to-Video Generation without Text-Video Data." arXiv,2022.

[9] H et al. "Grounded Decoding: Guiding Text Generation with Grounded Models for Robot Control." arXiv:2303.00855, 2023.

[10] Hafner et al. "Mastering diverse domains through world models." arXiv, 2023.

---

### Comment · Area_Chair_hqsS · 2023-08-18
**Please read and respond to rebuttal**

@all reviewers, please read the author's rebuttal and respond accordingly. Thanks!

Also, the authors have sent the following link to code: https://drive.google.com/drive/folders/1nEFmbfzZuVutekbipBTzhNn0COe0BKsj?usp=sharing

---

### Decision · Program_Chairs · 2023-09-21

**Decision:**

Accept (poster)

**Comment:**

The paper presents HiP, which uses several foundation models to produce robot policies. It uses an LLM to plan, a video model to create dynamically feasible trajectory, and an inverse dynamics model to output actions. HiP also enforces consistency via iterative refinement. The results are shown on robotic manipulation tasks to outperform baselines.

Generally, through the rebuttal process the authors have improved the paper, particularly through additional baselines and experiments, which the authors thoroughly implemented. As such, the reviewers have improved their score towards accepting the paper. The negative review is from 3JTC, who hasn’t responded. Overall, the paper presents a scalable, timely, and interesting approach and thoroughly ablates and compares HiP. The experiments are compelling, with the exception of no real-world experiments.

Based on the improvements and overall result, I recommend accepting the paper.

I do agree with reviewer 3JTC’s remaining point of the title does not clearly show the contribution and novelty of the paper. The authors should change it for the camera ready, perhaps to indicate the use of the specific models (language model for tasks, video model for dynamics, and inverse dynamics model for actions) and to indicate the use-case presented: robotic manipulation or visual-motor control. Without specifying a specific title, the authors should consider these guidelines.